# Businesses' Role in the Fulfillment of the 2030 Agenda: A Bibliometric Analysis

**María Garrido-Ruso** [1,*] **, Beatriz Aibar-Guzmán** [1] **and Albertina Paula Monteiro** [2]

1   Department of Financial Economics and Accounting, University of Santiago de Compostela, 15782 Santiago de Compostela, Spain; beatriz.aibar@usc.es
2   Porto Accounting and Business School, Polytechnic of Porto, 4465-004 Matosinhos, Portugal; amonteiro@iscap.ipp.pt
*   Correspondence: mariagarrido.ruso@usc.es

**Abstract:** Companies worldwide can play a fundamental role in the fulfillment of the 2030 Agenda. This paper aims to determine the scope of the existing literature about the role that organizations play in contributing to the advancement of Sustainable Development Goals (SDGs). A bibliometric analysis is conducted considering the papers specifically focused on SDGs and businesses published from 2015 to 2021 in journals indexed in the Scopus database. The analysis shows that approximately 80% of the studies on this topic have been published in the last three years. Moreover, only one journal (*Sustainability*) has published more than the 50% of the publications on the subject. The final sample is divided into 11 clusters that analyze different perspectives within the same research topic, and, in all these clusters, practically all of the papers have been published in the last two years, which confirms that this issue is increasing its presence in the academic world. This work extends the existing research on the subject, taking into account the publications of the last year, so it is an update on this "hot topic". Moreover, it contributes to providing a reference frame of the state of the art of this research topic and can orientate researchers in the development of future studies

**Keywords:** Agenda 2030; sustainable development goals; business; private sector





## 1. Introduction

The world is changing and the impact that activities have on our planet is provoking more and more negative consequences, which has meant that the main institutions worldwide have made a global commitment necessary to stop this deterioration. That is why the United Nations [1] proposed to continue the world's economic development in a sustainable way [2] and established the Millennium Development Goals (MDGs) in September 2000 [1–3].

Fifteen years later, more ambitious goals were set to continue on the path of the MDGs, and the UN defined the 2030 Agenda and their 17 Sustainable Development Goals (SDGs) with the intention of achieving a better world [4–6]. The main difference between the two proposals is that the SDGs are more global and involve not only government institutions, but also any type of private organization, so that companies can acquire a fundamental role, from this moment, to contribute to sustainable development [7].

The main difference between the ODM and the ODS is that the latter considers that any type of company can provide solutions for greater sustainability. It is about creating value and avoiding damage to the environment by carrying out their activity as little as possible, based on sustainable business models [8]. Therefore, companies worldwide can play a fundamental role in the fulfillment of the 2030 Agenda [9].

During the last few years, a stream of research about the implications of the SDGs for business strategies started. This study aims to analyze the state of the art in such research with the intention of determining the main issues surrounding this topic. The methodology followed was a bibliometric analysis of papers focused on the role that companies have

in contributing to the fulfillment of the SDGs published from 2015 to 2021 in journals indexed on the Scopus database. We evaluated the temporal evolution of publications, the number of publications per journal and year, the number of publications per country, and the number of publications by author. This study contributes to the SDG literature with a very complete analysis of the existing research on the role that businesses can play in achieving the SDGs and provides a clear summary of the subject. Consequently, we provide a systematization of the extant research on this subject that allows the identification of knowledge flows, active research topics, and lead authors, among other issues. Thus, this study's findings depict the current status of the research on the role of businesses in the fulfillment of SDGs and provide a frame of reference that could guide researchers regarding the direction of future studies on this subject.

The rest of this paper is structured as follows: after this introduction, the next section contextualizes the SDGs and explains the role that companies can play in achieving them. Section 3 contains the empirical framework of the analysis and, consequently, in Section 4, the main findings are presented. Finally, Section 5 presents the main conclusions of the study, the implications of the findings, and some limitations and topics for future researchers.

## 2. Theory

### 2.1. Sustainable Development Goals

The SDGs were defined in September 2015 by the United Nations at the United Nations General Assembly in New York [10]. The highest authorities of more than 150 countries met to approve the 2030 Agenda for Sustainable Development [2]. Under the name "Transforming Our World: The 2030 Agenda for Sustainable Development", a number of proposals were defined, and the 193 countries that are members of the UN committed to fulfilling this plan [2,11,12].

The main objective of this meeting was to achieve a commitment to a better world; therefore, the 2030 Agenda included 169 targets and 261 indicators, grouped into 17 SDGs (Figure 1), with the aim of improving our environment by guaranteeing sustainable development in all possible areas (social, economic, and environmental) [1–3,13–16]. Specifically, the 17 objectives are: (1) no poverty, (2) zero hunger, (3) good health and well-being, (4) quality education, (5) gender equality, (6) clean water and sanitation, (7) affordable and clean energy, (8) decent work and economic growth, (9) industry, innovation, and infrastructure, (10) reduced inequalities, (11) sustainable cities and communities, (12) responsible consumption and production, (13) climate action, (14) life below water, (15) life on land, (16) peace, justice, and strong institutions, and (17) partnerships for the goals.

As we can see, most of the SDGs deal with issues as important and serious as human rights, and they cover actions for eradicating inequalities (e.g., poverty, hunger, health, or education) and the bad habits that exist today on our planet, proposing a sustainable way of living [13,17–19]. The exception is SDG 17 "Partnerships for the goals"—this objective is the only one that, instead of establishing a purpose to be achieved, indicates the procedure to be followed to meet the other objectives. Compliance with the SDGs is not just a matter for the public institutions of each country—it is necessary that all agents align themselves to achieving a better world. This means that not only should governments implement policies and actions to meet these goals by 2030, but private organizations should also be involved in these objectives [6,20].

Moreover, it is necessary to highlight the correlation that exists between the objectives set by the UN. This means that any defined plan to improve one of the 17 objectives will have an impact on the others, so organizations should consider these goals as a whole [12,21,22]. They should not focus on one specific objective, since the interrelationship that exists between the 17 should lead to the design of a joint action plan to have an impact on several of these objectives [12].

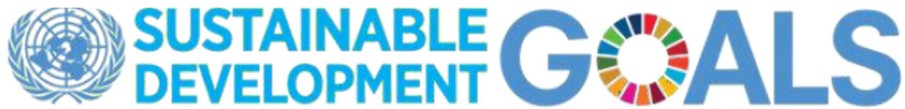

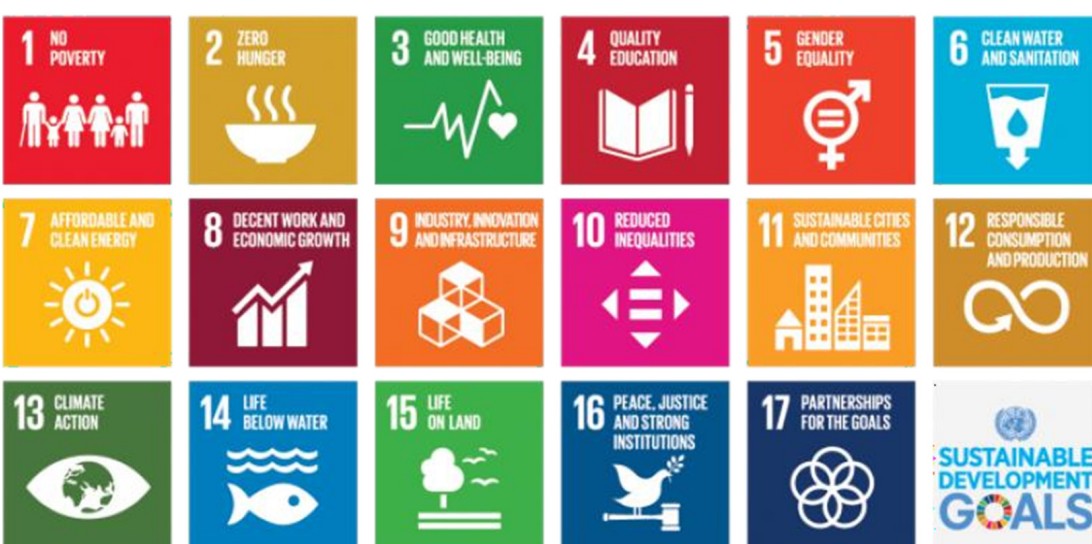

**Figure 1.** Source: https://www.un.org/es/sustainable-development-goals (accessed on 23 May 2022).

If the deadlines established by the UN are met, within 8 years, these 17 objectives should have been achieved. That is why, at the beginning of the 2020–2030 decade, the leaders involved in this mission defined a plan to "accelerate the compliance with the SDGs by 2030" [6] (p. 61). However, no one could imagine that this plan would be threatened by the COVID-19 pandemic [17,23]. In the year 2020, an unthinkable situation in the 21st century caused economic life to remain stagnant and the priority of governments to be managing the health situation that was being experienced. Consequently, the 2030 Agenda became something that remained in the background [23]. Practically, all of the SDGs have been affected by the COVID-19 pandemic that we have been experiencing since 2020, but SDG 3 has been affected in a more pronounced way [17].

*2.2. Business and SDGs*

As SDG 17 establishes, the SDGs should be achieved by partnerships [24]. This means that this is not an issue that only affects public institutions or governments—companies are a key element in achieving the SDGs [14,18,25–28]. The SDGs are of such magnitude that it is not enough for one actor to commit to them; commitments of businesses, governments, non-governmental organizations, and stakeholders are needed [1,2,28,29].

The United Nations defend the key role that organizations play in this context. Specifically, the 2030 Agenda states that "we acknowledge the diversity of the private sector, ranging from micro-enterprises to cooperatives to multinationals. We call upon all businesses to apply their creativity and innovation to solving sustainable development challenges" [1] (p. 29).

Therefore, the question that companies must ask themselves is how to collaborate with SDGs and how to incorporate this into their strategy [3,26,28]. Organizations must design their business plans from a more sustainable perspective considering two premises: harm the SDGs as little as possible and implement actions to help achieve those goals (e.g., save energy, reduce emissions, circular economy, etc.) [6,15,26,29–31]. This is the new challenge for businesses to not just maximize their benefits; now, they must do so in a sustainable way and collaborate with the environment that surrounds us [28,32].

We must consider that, these days, the economic objective is not the only factor that moves an organization. With all of the inequalities and problems mentioned above, it is essential that the commitment of companies to the SDGs has fundamental importance within the organizations, because it is a key tool to be competitive in the long-term [30].

Until recently, the commitment of companies to society was based on specific actions, such as donations or participation in some social activity, but this is not enough [30]. This mission involves a huge complexity for existing companies, since it is very difficult to change the general perspective of work; for startups, or for new companies or entrepreneurs, the idea would be to create a concept from scratch, based on the sustainable economy [4,26]. Moreover, this is an opportunity for businesses to work in a sustainable way, showing their stakeholders their commitment to Corporate Social Responsibility (CSR) activities [17,18,26,33]. At the beginning of the 2030 Agenda, and after a survey carried out at a company level, "more than 70% of global corporations plan to incorporate SDGs into their business and more than 40% plan to include SDGs in their business strategy within five years" [30] (p. 202).

This fundamental role that companies are playing in achieving the SDGs is reflected in the academic field. A stream of research is investigating the relationship between business and the SDGs. It is a relatively new topic, considering that the SDGs were defined in 2015. The main question is how companies can incorporate the SDGs within their corporate strategy [34,35]. Khaled et al. demonstrated the importance of this topic, affirming that "it is crucial to explore potential frameworks that would guide companies on how they can align their strategies as well as measure and communicate their contribution to the SDGs" [14] (p. 1). There are many questions about the relationship between the SDGs and business performance (e.g., if they prioritize SDGs or focus on a global perspective, if they elaborate on SDGs reports, if these activities have economic advantages for companies, and how the SDGs are perceived by their stakeholders) [26,29,34,36].

Taking into account the fundamental role of companies in contributing to the SDGs, the UN Global Compact, the GRI, and the World Business Council for Sustainable Development elaborated a document, the SDG Compass, to help businesses to include the SDGs in their plans [37]. This guide explains to companies how to include the SDGs in their strategy and how they should communicate it so that this information reaches their stakeholders [36,38]. Specifically, the SDG Compass defines five steps: (1) Understanding the SDGs; (2) Defining priorities; (3) Setting goals; (4) Integrating; and (5) Reporting.

## 3. Data and Methods

### 3.1. Sample Selection

With the objective of answering the research question, we conducted a bibliometric analysis. The first step in this process was to select the papers that we were going to analyze. First of all, we started a literature review focused on the topic and, after reviewing a considerable number of articles related to the topic, we defined our search criteria:

1.  As we explained before, the SDGs were defined in 2015 by the United Nations, so we started our search that year and we covered until the year 2021 to be able to analyze all of the possible complete years from its definition to the present;
2.  Papers were selected from Scopus, because it includes a wide range of studies about this topic, has more journals indexed than the Web of Science, and is a very common tool used for bibliometric studies [39,40];
3.  We focus our search on journal articles, rejecting other results, such as conferences or books chapters, among others;
4.  To obtain a more complete and interdisciplinary result, no filter referring to the different areas of knowledge was included;
5.  The articles should be written in English;
6.  Our search criteria were: "Title, keywords, or abstract".

Following these steps, we introduced into the Scopus database the following search:
(TITLE ("SDG") OR TITLE ("Sustainable Development Goal") OR TITLE ("SDG*") OR TITLE ("Sustainable Development Goal*") OR TITLE ("GLOBAL AGENDA") OR TITLE ("2030 agenda") OR TITLE ("Agenda 2030") OR TITLE ("SUSTAINABLE DEVELOPMENT AGENDA") AND KEY ("SDG") OR KEY ("SUSTAINABLE DEVELOPMENT GOAL") AND ABS ("organisation") OR ABS ("firm") OR ABS ("corporat*") OR ABS ("com

pany") OR ABS ("business") OR ABS ("ENTERPRISE") OR ABS ("PRIVATE SECTOR"))
AND (LIMIT-TO (SRCTYPE, "j")) AND (LIMIT-TO (DOCTYPE, "ar")) AND (LIMIT-TO
(PUBYEAR, 2021) OR LIMIT-TO (PUBYEAR, 2020) OR LIMIT-TO (PUBYEAR, 2019) OR
LIMIT-TO (PUBYEAR, 2018) OR LIMIT-TO (PUBYEAR, 2017) OR LIMIT-TO (PUBYEAR,
2016) OR LIMIT-TO (PUBYEAR, 2015)) AND (LIMIT-TO (LANGUAGE, "English")).

This search returned 543 empirical and non-empirical studies. Once we obtained these
results, we firstly read the abstracts of all of the articles to check if they really dealt with the
topic that we wanted to investigate.

After this first impression, in which some invalid results were already eliminated, we
started the next step, in which each of the authors separately read and analyzed the papers,
summarizing their main characteristics, and, subsequently, the results were compared. In
this analysis, papers focused on public organizations or those conducted in an academic
setting were eliminated.

Finally, 196 papers were identified. Figure 2 summarizes the steps taken to obtain the
final sample.

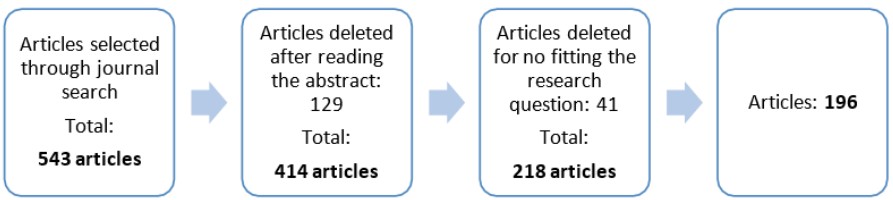

**Figure 2.** Search process.

### *3.2. Data Analysis and Procedure*

Once we obtained our final sample, we analyzed the data using the software VOSviewer,
specifically version 1.6.18. It was created by Nees Jan van Eck and Ludo Waltman CWTS
Leiden University, Leiden, The Netherlands, with the objective of "creating maps based on
network data and visualizing and exploring maps" [41] (p. 3). This visualization software
package was adopted because of "its powerful user graphic-interface that can generate
maps to describe the connections of each analysis unit" [42] (p. 304).

Although there are other instruments that can be used for conducting literature reviews
(e.g., PRISMA-statement and SciMAT), we chose VOSviewer because it has been broadly
used in previous studies [43,44].

## 4. Findings

### *4.1. Scientific Production on the Role That Business Has in the Achievement of the SDGs*

Our analysis shows that we are facing an emerging issue in the academic world.
Although it is true that the SDGs were established in 2015, it was not until 2019 that
this topic began to gain strength in the literature. This evidence confirms that, initially,
compliance with the SDGs was considered the responsibility of public organizations, while,
in the last two years, the role of business has been promoted as a fundamental factor when
it comes to meeting these objectives.

Figure 3 shows the chronological evolution of the publications on the role that busi-
nesses play in the achievement of the SDGs since 2015. As can be seen, the research on this
topic actually started in 2016. with the work of Scheyvens et al. in the journal *Sustainable
Development*, and increased its presence in the literature from the year 2019 until today.
Most papers were published during the last two years, specifically 161, which is 82.14% of
the total published papers, so the trend of this topic is clearly increasing.

Table 1 reports the number of publications per journal. We selected journals with five or
more articles published about the topic, because the vast majority published four (1 journal),
three (3 journals), two (11 journals), or fewer (66 journals) studies. *Sustainability* is clearly
the journal with the highest number of publications, at 50 papers, with a great difference
from the second journal, which is the *Journal of Cleaner Production*, at 15 publications.

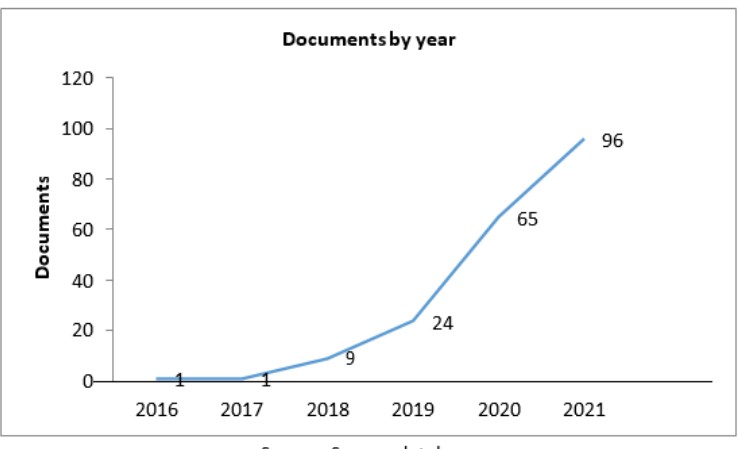

Source: Scopus database

**Figure 3.** Number of documents by year.

**Table 1.** Total number of publications per journal.

| Source | Documents |
|---|---|
| *Sustainability* | 50 |
| *Journal of Cleaner Production* | 15 |
| *Business Strategy and the Environment* | 7 |
| *Sustainable Development* | 6 |
| *Business Strategy and Development* | 5 |
| *Corporate Social Responsibility and Environmental Management* | 5 |
| *Worldwide Hospitality and Tourism Themes* | 5 |

Source: Scopus database.

Figure 4 provides the growth of sources attending to the number of articles published since 2015. The "*Sustainability*" journal has shown exponential growth in the number of articles published related to SDGs as the number of articles published in this journal in 2015 was 0, which has increased to 23 during the last year.

**Figure 4.** Documents per year by source.

The fact that we are working with such a novel topic in the academic world means that the authors who are dedicated to investigating this subject have not yet had time to publish a large number of articles on the topic. Figure 5 shows the authors that have published more than two papers about this topic. We can see that the maximum number of articles belonging to a researcher is four, a situation that García-Sánchez, van Tulder, and van Zanten share.

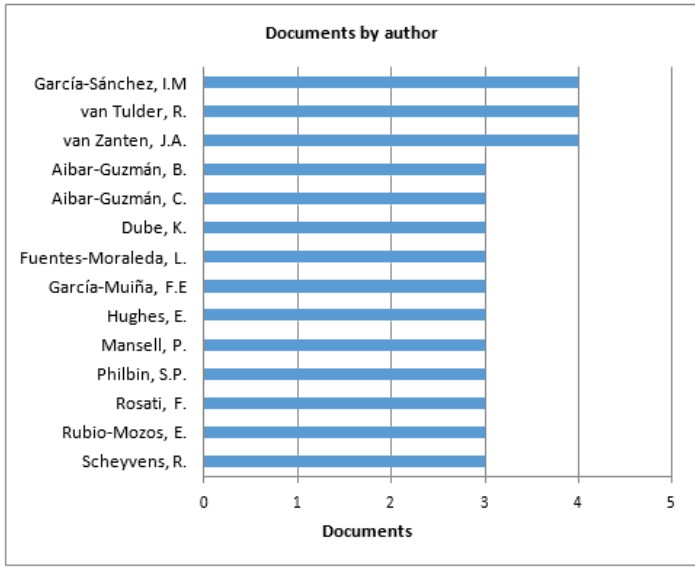

Source: Scopus database

**Figure 5.** Documents by author.

Figure 6 shows the distribution of the papers on the role that businesses have in the achievement of the SDGs. In total, we found more than 50 countries, and 29 with two or fewer publications. In Figure, 6 we included those that have three or more articles about this topic in Scopus. As can be seen, the country that has published the most papers on the role that business has in the achievement of the SDGs is Spain, with 30 articles, followed by the United Kingdom, with 27 papers.

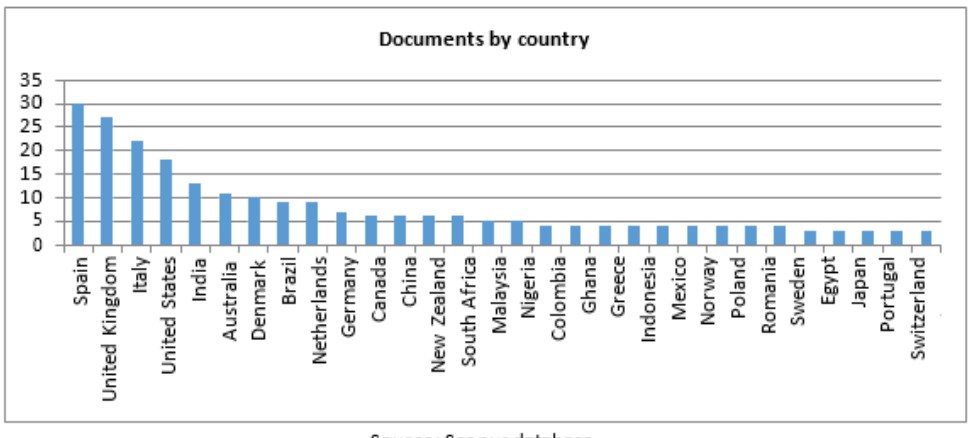

Source: Scopus database

**Figure 6.** Number of documents by country.

In Figure 6, we can see how the countries with the most published studies on this subject are developed countries, specifically, European countries (Spain, the United Kingdom, and Italy), followed by the United States. However, it should be noted that two countries of the BRICS, Brazil and China, are also among the top ten with a higher number of publications. This could mean that these countries are beginning to become involved in compliance with the SDGs, and their companies are already becoming aware of a more sustainable business model. Ali et al. reported on how BRICS countries are making efforts to engage their activities with the SDGs, but the main conclusion is that they are focusing only on some objectives, instead of covering them as a whole [45].

Finally, Table 2 shows the number of publications depending on the organization. To elaborate the table, we considered the most relevant organizations (those that have published three or more articles), since the vast majority have published two (35 institutions)

or one (111 organizations). As can be seen, the most productive universities are located in Europe. The University of Salamanca is the only one with five publications. Although the first places belong mostly to European universities, it is worth highlighting the second place of the University of Sao Paulo.

**Table 2.** Documents by organization.

| Organization | TP |
|---|---|
| University of Salamanca | 5 |
| University of Sao Paulo | 4 |
| University Rey Juan Carlos | 4 |
| University College London | 4 |
| Erasmus Universiteit Rotterdam | 4 |
| Parthenope University of Naples | 4 |
| University of Santiago de Compostela | 4 |
| Rotterdam School of Management, Erasmus University | 4 |
| University of Valencia | 3 |
| University of Oviedo | 3 |
| Vaal University of Technology | 3 |
| Massey University | 3 |
| Technical University of Denmark | 3 |
| University of Waterloo | 3 |
| University of the Aegan | 3 |
| Syddansk University | 3 |
| Copenhagen Business School | 3 |
| Uinversity Studi di Roma Tor Vergata | 3 |
| Sant'Anna Scuola Universitaria Superiore Pisa | 3 |
| London South Bank University | 3 |
| University of South Australia | 3 |
| Kwame Nkrumah University of Science and Technology | 3 |
| Bartlett Faculty of the Built Environment | 3 |

TP: total publications; Source: data collected from Scopus.

### 4.2. Research Subtopics

We conducted a bibliographic coupling analysis with the objective of identifying different research subtopics within the sample. This analysis was based on the idea that "the relatedness of items is determined based on the number of references they share" (vosViewer software, version 1.6.18; Nees Jan van Eck and Ludo Waltman, CWTS Leiden University, Leiden, The Netherlands). In this case, 13 out of the 196 publications did not have any kind of connection. Thus, the largest set of connected items was made up of 183 publications. Figure 7 shows the bibliographic coupling analysis of the publications on the role that business has in the achievement of the SDGs without considering the ones that are not connected to each other. Van Eck and Waltman claimed that the "clusters that are located close to each other tend to be strongly related in terms of citations, while clusters that are located further away from each other tend to be less strongly related" [46] (p. 1062).

Figure 7 shows the eleven clusters generated by the bibliographic coupling analysis.

VosViewer detected that three articles formed individual clusters and consequently, they are graphically represented in points completely separated from each other and from the main clusters, so we decided not to take them into account in this section. In addition, the initial result produced 11 clusters, but the reality was that the last two did not have enough links to be considered relevant. Therefore, below, we expose information on the first nine clusters in this analysis.

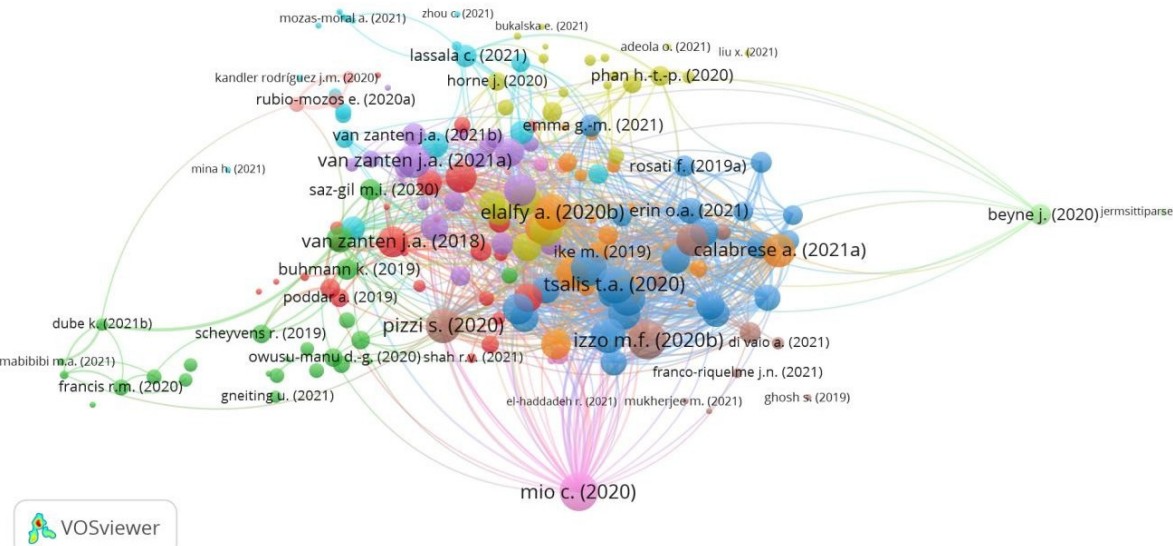

**Figure 7.** Bibliographic coupling analysis. Source: vosViewer.

*-Cluster 1 (colored red)—How businesses address the SDGs:* Twenty-five papers that analyzed the business contribution to the SDGs make up this cluster. The main topic of these works is how companies can perform to achieve Agenda 2030.

The articles with the highest number of links were those by Ordóñez-Ponce et al. (2021), Calabrese et al. (2021), van Zanten and van Tulder (2018), and Vildasen (2018). With regards to the articles' impacts, the paper with the highest number of citations, both in absolute and in relative terms, was that by Scheyvens et al. (2016) [47]. The following papers with higher academic impact were those by van Zanten and van Tulder (2018) [34] and Gunawan et al. (2020) [48]. Conversely, the papers with a lower number of citations were those by Bianchi (2021) [25], Andrian et al. (2021) [49], and Shah and Acharya (2021) [50]. These papers had no citations.

Almost all of the articles belonging to this cluster were written by multiple authors (20 papers), whereas there were 5 publications that were written by single authors. None of the authors in this group have published more than one article. Within this subtopic, the journal with more papers published was *Sustainability* (four papers), followed by *Marketing Intelligence and Planning* (three papers) and the *Journal of Cleaner Production and Sustainable Development*, with two papers each.

The first published article of this cluster dated from 2016, and the year with more publications was 2021; almost all of the papers belonging to this cluster were published in the last two years (18 papers), which is consistent with our statement above. Moreover, European countries were the most analyzed.

Table 3 shows the papers belonging to this cluster, their journal, the number of links between papers, the country or region of study, and their impact or influence measured by the total number of citations and the average number of citations per year from the date of publication (NIY) [51]. It should be noted that the last column reflects the "acceleration" of the impact in time weighting. Thus, under equal conditions of the date of publication, a greater NIY means greater academic interest in the paper.

**Table 3.** Cluster 1.

| RO | Author | Links | Journal | Country | Citations | NIY |
|---|---|---|---|---|---|---|
| 1 | Scheyvens et al. (2016) [47] | 60 | *Sustainable Development* | n.a. | 259 | 43.17 |
| 2 | van Zanten and van Tulder (2018) [34] | 103 | *Journal of International Business Policy* | Europe and North America | 152 | 38 |
| 3 | Gunawan et al. (2020) [48] | 11 | *Journal of Cleaner Production* | Indonesia | 35 | 17.5 |
| 4 | Avrampou et al. (2019) [52] | 73 | *Sustainable Development* | Europe | 47 | 15.67 |
| 5 | Tabares (2021) [53] | 33 | *Journal of Cleaner Production* | Colombia | 13 | 13 |
| 6 | Calabrese et al. (2021) [26] | 104 | *Technological Forecasting and Social Change* | International | 10 | 10 |
| 7 | Ali et al. (2018) [45] | 38 | *Sustainability* (Switzerland) | BRICS | 39 | 9.75 |
| 8 | Poddar et al. (2019) [54] | 72 | *Corporate Social Responsibility and Environmental Management* | India | 29 | 9.67 |
| 9 | Palakshappa and Dodds (2021) [55] | 15 | *Marketing Intelligence and Planning* | Canada and New Zealand | 9 | 9 |
| 10 | Yu et al. (2020) [56] | 74 | *Sustainability* (Switzerland) | China | 18 | 9 |
| 11 | Goyal et al. (2021) [57] | 13 | *Qualitative Research in Organizations and Management: An International Journal* | India | 8 | 8 |
| 12 | Günzel-Jensen et al. (2020) [58] | 42 | *Journal of Business Venturing Insights* | Germany | 14 | 7 |
| 13 | Lopez (2020) [59] | 35 | *Marketing Intelligence and Planning* | Spain | 12 | 6 |
| 14 | Jonsdottir et al. (2021) [35] | 47 | *Sustainability* (Switzerland) | Iceland | 5 | 5 |
| 15 | Ordonez-Ponce et al. (2021) [18] | 121 | *Sustainability Accounting, Management and Policy Journal* | International | 4 | 4 |
| 16 | Krantz and Gustafsson (2021) [60] | 48 | *Journal of Environmental Planning and Management* | Swedish | 4 | 4 |
| 17 | Hepner et al. (2021) [61] | 8 | *Marketing Intelligence and Planning* | International | 4 | 4 |
| 18 | Escher and Brzustewicz (2020) [62] | 47 | *Sustainability* (Switzerland) | Poland | 8 | 4 |
| 19 | Bello and Othman (2020) [63] | 4 | *International Journal of Educational Management* | Nigeria | 8 | 4 |
| 20 | Vildåsen (2018) [64] | 92 | *Business Strategy and Development* | Finland | 10 | 2.5 |

**Table 3.** *Cont.*

| RO | Author | Links | Journal | Country | Citations | NIY |
|---|---|---|---|---|---|---|
| 21 | Díaz-Perdomo et al. (2021) [65] | 15 | *Frontiers in Psychology* | Spain | 2 | 2 |
| 22 | Antonaras (2018) [66] | 8 | *Cyprus Review* | Cyprus | 3 | 0.75 |
| 23 | Bianchi (2021) [25] | 65 | *Sustainability* (Switzerland) | n.a. | 0 | 0 |
| 24 | Andrian et al. (2021) [49] | 39 | *Review of International Geographical Education Online* | Indonesia | 0 | 0 |
| 25 | Shah and Acharya (2021) [50] | 24 | *Ecology, Environment and Conservation* | n.a. | 0 | 0 |

RO: ranking order; NIY: normalized citations per year; Source: Scopus.

*-Cluster 2 (colored green)—Benefits arising from SDG engagement:* Twenty-four papers that have analyzed how companies can benefit from the process of aligning their activities to the SDGs make up this cluster.

The articles with the highest number of links were those by Imaz and Eizagirre (2020), Buhmann et al. (2019), Owusu-Manu et al. (2020), and Saz-Gil et al. (2020). With regards to the articles' impacts, the paper with the highest number of citations in relative terms was that by Endl et al. (2021) [67], whereas the paper with more total citations was that by Monteiro et al. (2019) [68]. Conversely, the papers with a lower number of citations were those by Wankel (2021) [69], which had no citations, Jones et al. (2018) [70], and Francis and Nair (2020) [71].

Almost all the articles belonging to this cluster were written by multiple authors (20 papers), whereas there were four publications by single authors. The authors with a higher number of publications were Dube, K., with three papers, followed by Comfort, D., Hughes, E., Jones, P., Nair, V., and Scheyvens, R., with two papers each. Within this subtopic, the journal with more papers published was *Sustainability* (six papers), followed by *Worldwide Hospitality and Tourism Themes* (three papers). The rest of the journals in this cluster have published a single article.

The first published article of this cluster dated from 2018, and the years with more publications were 2020 and 2021, with ten articles each year. Moreover, this cluster included papers analyzing different regions around the world

Table 4 shows the papers belonging to this cluster, their journal, the number of links between papers, the country or region of study, and their impact or influence measured by the total number of citations and the average number of citations per year from the date of publication (NIY) [51]. It should be noted that the last column reflects the "acceleration" of the impact in time weighting. Thus, under equal conditions of the date of publication, the greater the NIY, the greater the academic interest in the paper.

**Table 4.** Cluster 2.

| RO | Author | Links | Journal | Country | Citations | NIY |
|---|---|---|---|---|---|---|
| 1 | Endl et al. (2021) [67] | 12 | *Resources Policy* | International | 24 | 24 |
| 2 | Monteiro et al. (2019) [68] | 2 | *Journal of Cleaner Production* | n.a. | 58 | 19.33 |
| 3 | Scheyvens and Hughes (2019) [72] | 49 | *Journal of Sustainable Tourism* | Fiji | 57 | 19 |

**Table 4.** *Cont.*

| RO | Author | Links | Journal | Country | Citations | NIY |
|---|---|---|---|---|---|---|
| 4 | KC et al. (2021) [73] | 49 | *Tourism Management Perspectives* | Nepal | 11 | 11 |
| 5 | Scheyvens et al. (2021) [74] | 60 | *Annals of Tourism Research* | Fiji, Australia, New Zealand | 10 | 10 |
| 6 | Kumi et al. (2020) [75] | 54 | *Extractive Industries and Society* | Ghana | 18 | 9 |
| 7 | Buhmann et al. (2019) [76] | 74 | *Corporate Governance* (Bingley) | n.a. | 26 | 8.67 |
| 8 | Dube and Nhamo (2021) [77] | 13 | *GeoJournal* | South Africa | 7 | 7 |
| 9 | Olwig (2021) [78] | 66 | *World Development* | Denmark | 4 | 4 |
| 10 | Owusu-Manu et al. (2020) [79] | 70 | *Journal of Engineering, Design and Technology* | Ghana | 8 | 4 |
| 11 | García-Sánchez et al. (2020) [38] | 55 | *Sustainability* (Switzerland) | Spain | 8 | 4 |
| 12 | Imaz and Eizagirre (2020) [80] | 98 | *Sustainability* (Switzerland) | n.a. | 7 | 3.5 |
| 13 | Saz-Gil et al. (2020) [81] | 70 | *Sustainability* (Switzerland) | n.a. | 7 | 3.5 |
| 14 | Consolandi et al. (2020) [82] | 68 | *Organization and Environment* | United States | 7 | 3.5 |
| 15 | Olofsson and Mark-Herbert (2020) [83] | 63 | *Sustainability* (Switzerland) | Swedish | 4 | 2 |
| 16 | Milwood (2020) [84] | 50 | *Worldwide Hospitality and Tourism Themes* | Caribe | 3 | 1.5 |
| 17 | Nair and McLeod (2020) [85] | 2 | *Worldwide Hospitality and Tourism Themes* | Caribe | 3 | 1.5 |
| 18 | Francis and Nair (2020) [71] | 51 | *Worldwide Hospitality and Tourism Themes* | Bahamas | 2 | 1 |
| 19 | Gneiting and Mhlanga (2021) [86] | 46 | *Development in Practice* | - | 1 | 1 |
| 20 | Dube (2021) [27] | 7 | *Sustainability* (Switzerland) | Botswana and Zimbabwe | 1 | 1 |
| 21 | Mabibibi et al. (2021) [87] | 6 | *Sustainability* (Switzerland) | South Africa | 1 | 1 |
| 22 | Jones and Comfort (2021) [88] | 4 | *Property Management* | United Kingdom | 1 | 1 |
| 23 | Jones et al. (2018) [70] | 51 | *World Review of Entrepreneurship, Management and Sustainable Development* | United Kingdom | 3 | 0.75 |
| 24 | Wankel (2021) [69] | 55 | *IBIMA Business Review* | n.a. | 0 | 0 |

RO: ranking order; NIY: normalized citations per year; Source: Scopus.

-*Cluster 3 (colored blue)—SDG reporting. Disclosure level and determinants:* Twenty-four papers that have analyzed SDG reporting make up this cluster. This practice is essential for stakeholders to be aware of the involvement that companies have in the 2030 Agenda. The relevance of the SDG disclosure is such that, in this analysis, we found three clusters that dealt with this issue, but from different perspectives. Therefore, the articles belonging to this subtopic had a closer link with those that formed clusters 7 and 8.

The articles with the highest number of links were those by Tsalis et al. (2020), Sardianou et al. (2020), Pizzi et al. (2021), Battaglia et al. (2020) and Izzo et al. (2020). With regards to the articles' impact, the paper with the highest number of citations in relative terms was that by Pizzi et al. (2021) [89], whereas the paper with more total citations was that by Rosati and Faria (2019) [9]. Conversely, the paper with a lower number of citations was that by Liu et al. (2021) [15], with no citations.

Almost all of the articles belonging to this cluster were written by multiple authors (20 papers), whereas there was only one paper with a single author. The authors with a higher number of publications were Aibar-Guzmán, B., Aibar-Guzmán, C., and García-Sánchez, I.M., with three papers each, followed by García-Meca, E., Nikolaou, I., Rodríguez-Ariza, L., and Rosati, F., with two papers each. Within this subtopic, the journal with more papers published was *Sustainability* (seven papers), followed by the *Journal of Cleaner Production* (four papers) and *Corporate Social Responsibility and Environmental Management* (three papers). The first published article of this cluster dated from 2019, and the year with more publications was 2021, with 15 articles published during that year. Moreover, European countries were the most analyzed.

Table 5 shows the papers belonging to this cluster, their journal, the number of links between papers, the country or region of study, and their impact or influence measured by the total number of citations and the average number of citations per year from the date of publication (NIY) [51]. It should be noted that the last column reflects the "acceleration" of the impact in time weighting. Thus, under equal conditions of the date of publication, the greater the NIY, the greater the academic interest in the paper.

**Table 5.** Cluster 3.

| RO | Author | Links | Journal | Country | Citations | NIY |
|----|--------|-------|---------|---------|-----------|-----|
| 1 | Pizzi et al. (2021) [89] | 108 | *Business Strategy and the Environment* | Italy | 49 | 49 |
| 2 | Tsalis et al. (2020) [90] | 117 | *Corporate Social Responsibility and Environmental Management* | n.a. | 81 | 40.5 |
| 3 | Rosati and Faria (2019) [9] | 63 | *Corporate Social Responsibility and Environmental Management* | International | 99 | 33 |
| 4 | Curtó-Pagès et al. (2021) [91] | 100 | *Sustainability* (Switzerland) | Spain | 14 | 14 |
| 5 | Fonseca and Carvalho (2019) [92] | 80 | *Sustainability* (Switzerland) | Portugal | 39 | 13 |
| 6 | García-Meca and Martínez-Ferreiro. (2021) [93] | 70 | *Journal of Cleaner Production* | Europe | 11 | 11 |
| 7 | García-Sánchez et al. (2020) [36] | 63 | *Journal of Cleaner Production* | Spain | 21 | 10.5 |
| 8 | Diaz-Sarachaga (2021) [29] | 81 | *Corporate Social Responsibility and Environmental Management* | Spain | 10 | 10 |

**Table 5.** *Cont.*

| RO | Author | Links | Journal | Country | Citations | NIY |
|----|--------|-------|---------|---------|-----------|-----|
| 9 | Di Vaio and Varriale (2020) [94] | 81 | *Journal of Cleaner Production* | Italy | 17 | 8.5 |
| 10 | García-Sánchez et al. (2019) [95] | 80 | *Business Strategy and the Environment* | Spain | 25 | 8.33 |
| 11 | Gallego-Sosa et al. (2021) [96] | 61 | *Sustainability* (Switzerland) | Europe | 7 | 7 |
| 12 | Erin and Bamigboye (2021) [97] | 83 | *Journal of Accounting and Organizational Change* | Africa | 7 | 7 |
| 13 | Martínez-Ferrero and García-Meca (2020) [98] | 81 | *Sustainable Development* | Europe | 13 | 6.5 |
| 14 | Khaled et al. (2021) [14] | 86 | *Journal of Cleaner Production* | International | 5 | 5 |
| 15 | Nishitani et al. (2021) [99] | 97 | *Journal of Environmental Management* | Vietnam | 3 | 3 |
| 16 | Haywood and Boihang (2021) [100] | 93 | *Development Southern Africa* | South Africa | 3 | 3 |
| 17 | Izzo et al. (2020) [101] | 107 | *Sustainability* (Switzerland) | Europe | 6 | 3 |
| 18 | Sardianou et al. (2020) [102] | 116 | *Sustainable Production and Consumption* | Europe | 5 | 2.5 |
| 19 | García-Sánchez et al. (2021) [103] | 70 | *Sustainable Development* | Spain | 2 | 2 |
| 20 | Jun and Kim (2021) [104] | 64 | *Sustainability* (Switzerland) | South Korea | 2 | 2 |
| 21 | Battaglia et al. (2020) [105] | 107 | *Business Strategy and Development* | Italy | 2 | 1 |
| 22 | Sekarlangit and Wardhani (2021) [106] | 56 | *Sustainability* (Switzerland) | Southeast Asia | 1 | 1 |
| 23 | Kazemikhasragh et al. (2021) [107] | 49 | *International Journal of Technology Management and Sustainable Development* | Asia and Africa | 1 | 1 |
| 24 | Liu et al. (2021) [15] | 58 | *Sustainability* (Switzerland) | Colombia and Egypt | 0 | 0 |

RO: ranking order; NIY: normalized citations per year; Source: Scopus.

*-Cluster 4 (colored yellow)—Corporate sustainability and SDGs:* Twenty-three papers that analyzed the relationship between corporate sustainability and Agenda 2030 make up this cluster.

The articles with the highest number of links were those by Modgil et al. (2020), van der Waal and Thijssens (2020), and Claro and Esteves (2020). With regards to the articles' impact, the paper with the highest number of citations in relative terms was that by van der Waal and Thijssens (2020) [108], whereas the paper with more total citations was that by Chams and García-Blandón (2019) [109]. Conversely, there were five papers with no citations.

All the articles belonging to this cluster were written by multiple authors. The author with a higher number of publications was Phan H.-T.-P., with two papers. Within this subtopic, the journal with more papers published was *Sustainability* (five papers), followed by the *Journal of Cleaner Production* (two papers), while the other journal had published one article each. The first published article of this cluster dated from 2019, and the year with more publications was 2020, with 11 papers published during that year, followed by 2021, with ten. Moreover, this cluster included papers analyzing different regions around the world.

Table 6 shows the papers belonging to this cluster, their journal, the number of links between papers, the country or region of study, and their impact or influence measured by the total number of citations and the average number of citations per year from the date of publication (NIY) [51]. It should be noted that the last column reflects the "acceleration" of the impact in time weighting. Thus, under equal conditions of the date of publication, the greater the NIY, the greater the academic interest in the paper.

**Table 6.** Cluster 4.

| RO | Author | Links | Journal | Country | Citations | NIY |
|---|---|---|---|---|---|---|
| 1 | van der Waal and Thijssens (2020) [108] | 109 | *Journal of Cleaner Production* | International | 64 | 32 |
| 2 | Chams and García-Blandón (2019) [109] | 20 | *Resources, Conservation and Recycling* | n.a. | 91 | 30.33 |
| 3 | Horne et al. (2020) [110] | 65 | *Journal of Cleaner Production* | Germany | 54 | 27 |
| 4 | Ilyas et al. (2020) [111] | 26 | *Environmental Science and Pollution Research* | Pakistan | 42 | 21 |
| 5 | Centobelli et al. (2020) [112] | 18 | *Technological Forecasting and Social Change* | Europe | 42 | 21 |
| 6 | Muhmad and Muhamad (2021) [113] | 15 | *Journal of Sustainable Finance and Investment* | n.a. | 11 | 11 |
| 7 | Acuti et al. (2020) [114] | 55 | *Cities* | Italy and Japan | 19 | 9.5 |
| 8 | Modgil et al. (2020) [115] | 119 | *Production Planning and Control* | India | 18 | 9 |
| 9 | De Luca et al. (2020) [116] | 25 | *Sustainability* (Switzerland) | Italy | 12 | 6 |
| 10 | Jha and Rangarajan (2020) [117] | 98 | *Sustainable Development* | India | 10 | 5 |
| 11 | Santos and Silva Bastos (2021) [3] | 52 | *Social Responsibility Journal* | Portugal | 5 | 5 |
| 12 | Adeola et al. (2021) [118] | 8 | *World Journal of Entrepreneurship, Management and Sustainable Development* | - | 4 | 4 |
| 13 | Claro and Esteves (2020) [119] | 102 | *Marketing Intelligence and Planning* | - | 8 | 4 |
| 14 | Phan et al. (2020) [120] | 56 | *Sustainability* (Switzerland) | Italy | 8 | 4 |
| 15 | Liu et al. (2021) [32] | 3 | *Energy Economics* | China | 3 | 3 |
| 16 | Chaurasia et al. (2021) [121] | 21 | *Decision Sciences* | n.a. | 2 | 2 |

**Table 6.** *Cont.*

| RO | Author | Links | Journal | Country | Citations | NIY |
|----|--------|-------|---------|---------|-----------|-----|
| 17 | Bhaskar and Kumar (2019) [122] | 54 | *Journal of Indian Business Research* | n.a. | 5 | 1.67 |
| 18 | Singh and Rahman (2021) [123] | 93 | *Cogent Business and Management* | India | 0 | 0 |
| 19 | Gallardo-Vázquez et al. (2021) [124] | 66 | *Sustainability* (Switzerland) | Spain | 0 | 0 |
| 20 | Socoliuc et al. (2020) [125] | 23 | *Polish Journal of Environmental Studies* | Rumania | 0 | 0 |
| 21 | Yu and Kuo (2021) [126] | 20 | *Sustainability* (Switzerland) | China | 1 | 1 |
| 22 | Nobrega et al. (2021) [127] | 7 | *Sustainability* (Switzerland) | Brazil | 0 | 0 |
| 23 | Bukalska et al. (2021) [4] | 6 | *Energies* | Poland | 0 | 0 |

RO: ranking order; NIY: normalized citations per year; Source: Scopus.

*-Cluster 5—Business Interactions with the SDGs:* Twenty-two papers that analyzed the nexus between business and SDGs, raising questions as to whether the different characteristics of companies cause them to interact differently with SDGs, make up this cluster.

The articles with the highest number of links were those by Rygh et al. (2021), van Zanten and van Tulder (2021), and Javeed et al. (2021). With regards to the articles' impact, the paper with the highest number of citations in relative terms was that by van Zanten and van Tulder (2021) [128], whereas the paper with more total citations was that by Fleming et al. (2017) [129]. Conversely, there were three papers with no citations.

Almost all of the articles belonging to this cluster were written by multiple authors (17 papers), whereas there were five publications by single authors. The authors with a higher number of publications were van Tulder, R., and van Zanten, J.A., with three papers each. Within this subtopic, the journal with more papers published was *Sustainability* (four papers), followed by *Business Strategy and Development* (three papers), *Business Strategy and the Environment* (two papers), and *Corporate Governance* (two papers). The first published article of this cluster dated from 2017, and the year with more publications was 2021, with 13 papers published during that year.

Table 7 shows the papers belonging to this cluster, their journal, the number of links between papers, the country or region of study, and their impact or influence measured by the total number of citations and the average number of citations per year from the date of publication (NIY) [51]. It should be noted that the last column reflects the "acceleration" of the impact in time weighting. Thus, under equal conditions of the date of publication, the greater the NIY, the greater the academic interest in the paper.

**Table 7.** Cluster 5.

| RO | Author | Links | Journal | Country | Citations | NIY |
|----|--------|-------|---------|---------|-----------|-----|
| 1 | van Zanten and van Tulder (2021) [130] | 102 | *Business Strategy and the Environment* | n.a. | 25 | 25 |
| 2 | van Zanten and van Tulder (2021) [131] | 66 | *International Journal of Sustainable Development and World Ecology* | n.a. | 25 | 25 |

**Table 7.** *Cont.*

| RO | Author | Links | Journal | Country | Citations | NIY |
|----|--------|-------|---------|---------|-----------|-----|
| 3 | van Zanten and van Tulder (2021) [28] | 76 | *Business Strategy and the Environment* | - | 14 | 14 |
| 4 | Sinkovics et al. (2021) [132] | 72 | *Multinational Business Review* | n.a. | 11 | 11 |
| 5 | Gutberlet (2021) [133] | 12 | *World Development* | Brazil | 9 | 9 |
| 6 | Pineda-Escobar (2019) [134] | 73 | *Corporate Governance* (Bingley) | Colombia | 25 | 8.33 |
| 7 | Fleming et al. (2017) [128] | 20 | *Marine Policy* | Australia | 38 | 7.6 |
| 8 | Liou and Rao-Nicholson (2021) [135] | 58 | *Journal of International Business Policy* | n.a. | 6 | 6 |
| 9 | Blagov and Petrova-Savchenko (2021) [136] | 57 | *Corporate Governance* (Bingley) | Russia | 5 | 5 |
| 10 | Dahlmann et al. (2019) [137] | 97 | *Anthropocene Review* | n.a. | 15 | 5 |
| 11 | Redman (2018) [138] | 49 | *Business Strategy and Development* | n.a. | 15 | 3.75 |
| 12 | Arnold (2018) [139] | 77 | *Business Strategy and Development* | International | 13 | 3.25 |
| 13 | Fei et al. (2021) [12] | 40 | *Sustainability* (Switzerland) | International | 3 | 3 |
| 14 | Malay and Aubinet (2021) [140] | 86 | *Ecological Economics* | Belgium | 2 | 2 |
| 15 | Buczacki et al. (2021) [141] | 37 | *Sustainability* (Switzerland) | n.a. | 2 | 2 |
| 16 | Lisowski et al. (2020) [142] | 67 | *Sustainability* (Switzerland) | International | 3 | 1.5 |
| 17 | Macellari et al. (2018) [143] | 58 | *Business Strategy and Development* | Italy | 5 | 1.25 |
| 18 | Khalique et al. (2021) [144] | 6 | *Australasian Accounting, Business and Finance Journal* | India | 1 | 1 |
| 19 | Fagerlin et al. (2019) [129] | 44 | *Sustainability* (Switzerland) | Japan | 1 | 0.33 |
| 20 | Rygh et al. (2021) [145] | 106 | *Critical Perspectives on International Business* | n.a. | 0 | 0 |
| 21 | Javeed et al. (2021) [146] | 99 | *Journal of Cultural Heritage Management and Sustainable Development* | Pakistan | 0 | 0 |
| 22 | Matteucci (2020) [147] | 13 | *Worldwide Hospitality and Tourism Themes* | international | 0 | 0 |

RO: ranking order; NIY: normalized citations per year; Source: Scopus.

-*Cluster 6 (colored light blue)—Performance, business model, and SDG measurement:* Nineteen papers that analyzed the relationship between performance and business model with SDGs in addition to articles dealing with SDG measurement make up this cluster.

The articles with the highest number of links were those by Ejarque and Campos (2020), Cordova and Celone (2019), and Nechita et al. (2020). With regards to the articles' impact, the paper with the highest number of citations, both in absolute and in relative terms, was that by Mina et al. (2021) [148]. The following papers with a higher academic impact were those by Lassala et al. (2021) [149] and Núñez et al. (2020) [150]. Conversely, the paper with the lowest number of citations was that by Kandler Rodríguez (2020) [151], with no citations.

Almost all of the articles belonging to this cluster were written by multiple authors (18 papers), whereas there was only 1 paper written by a single author. The authors with a higher number of publications were Mansell, P., and Philbin, S.P., with three papers each, followed by Mozas-Moral, A., Bernal-Jurado, E., Fernández-Uclés, D., and Medina-Viruel, M.J., with two papers each. Within this subtopic, the journal with more papers published was *Sustainability*. The first published article of this cluster dated from 2019, and the year with more publications was 2020, with 11 papers, and 2021, with 8 papers. Moreover, Spain was the most analyzed country in this cluster.

Table 8 shows the papers belonging to this cluster, their journal, the number of links between papers, the country or region of study, and their impact or influence measured by the total number of citations and the average number of citations per year from the date of publication (NIY) [51]. It should be noted that the last column reflects the "acceleration" of the impact in time weighting. Thus, under equal conditions of the date of publication, the greater the NIY, the greater the academic interest in the paper.

**Table 8.** Cluster 6.

| RO | Author | Links | Journal | Country | Citations | NIY |
|----|--------|-------|---------|---------|-----------|-----|
| 1 | Mina et al. (2021) [148] | 3 | *Journal of Cleaner Production* | - | 33 | 33 |
| 2 | Lassala et al. (2021) [149] | 67 | *Economic Research-Ekonomska Istrazivanja* | Spain | 13 | 13 |
| 3 | Núñez et al. (2020) [150] | 4 | *Sustainability* (Switzerland) | Spain | 14 | 7 |
| 4 | Cordova and Celone (2019) [152] | 89 | *Sustainability* (Switzerland) | n.a. | 15 | 5 |
| 5 | Mozas-Moral et al. (2020) [153] | 4 | *Sustainability* (Switzerland) | Spain | 8 | 4 |
| 6 | Mozas-Moral et al. (2021) [2] | 10 | *Technological Forecasting and Social Change* | Spain | 3 | 3 |
| 7 | Raiden and King (2021) [154] | 7 | *Resources, Conservation and Recycling* | England | 3 | 3 |
| 8 | Zhou and Etzkowitz (2021) [155] | 3 | *Sustainability* (Switzerland) | n.a. | 3 | 3 |
| 9 | Nechita et al. (2020) [156] | 84 | *Sustainability* (Switzerland) | East Europe | 6 | 3 |
| 10 | Mansell et al. (2020) [157] | 43 | *Sustainability* (Switzerland) | United Kingdom | 6 | 3 |
| 11 | Mansell et al. (2020) [158] | 25 | *Sustainability* (Switzerland) | United Kingdom | 5 | 2.5 |

**Table 8.** *Cont.*

| RO | Author | Links | Journal | Country | Citations | NIY |
|---|---|---|---|---|---|---|
| 12 | Mansell and Philbin (2020) [159] | 43 | *Journal of Modern Project Management* | n.a. | 4 | 2 |
| 13 | Jiménez et al. (2020) [160] | 67 | *Sustainability* (Switzerland) | Spain | 3 | 1.5 |
| 14 | Gambetta et al. (2021) [7] | 72 | *Journal of Legal, Ethical and Regulatory Issues* | - | 1 | 1 |
| 15 | Jiménez et al. (2021) [161] | 23 | *Sustainability* (Switzerland) | Spain | 1 | 1 |
| 16 | de la Casa and Caballero (2021) [162] | 4 | *CIRIEC-Espana Revista de Economia Publica, Social y Cooperativa* | Spain | 1 | 1 |
| 17 | Ejarque and Campos (2020) [163] | 101 | *Sustainability* (Switzerland) | Europe | 2 | 1 |
| 18 | Ionaşcu et al. (2020) [164] | 66 | *Sustainability* (Switzerland) | n.a. | 2 | 1 |
| 19 | Kandler Rodríguez (2020) [151] | 9 | *Worldwide Hospitality and Tourism Themes* | Costa Rica | 0 | 0 |

RO: ranking order; NIY: normalized citations per year; Source: Scopus.

*-Cluster 7 (colored orange)—SDG reporting. Its use with legitimation purpose:* Nineteen papers that analyzed the SDG reporting as a legitimation purpose make up this cluster. As mentioned above, the articles belonging to this subtopic had a closer link to those that formed clusters 3 and 8. The articles with the highest number of links were those by Elalfy et al. (2020), Elalfy et al. (2020), Calabrese et al. (2021), and van der Waal et al. (2021). With regards to the articles' impact, the paper with the highest number of citations in relative terms was that by van der Waal et al. (2021) [165], whereas the paper with more total citations was that by Ike et al. (2019) [166]. Conversely, there were two papers with no citations: Caldana et al. (2021) [167], and Galleli et al. (2021) [17].

Almost all of the articles belonging to this cluster were written by multiple authors (18 papers), while there was only 1 publication by a single author. The authors with a higher number of publications were ElAlfy, A., Khare, A., Krüger, C., LourenÇao, M., Pennabel, A.F., and Webber, O., with two papers each. Within this subtopic, the journals with more papers published were the *Journal of Cleaner Production* and *Sustainability*, with three papers each, followed by *Business Strategy and the Environment* (two papers). The first published article of this cluster dated from 2018, and the year with more publications was 2021, with twelve papers published during that year.

Table 9 shows the papers belonging to this cluster, their journal, the number of links between papers, the country or region of study, and their impact or influence measured by the total number of citations and the average number of citations per year from the date of publication (NIY) [51]. It should be noted that the last column reflects the "acceleration" of the impact in time weighting. Thus, under equal conditions of the date of publication, the greater the NIY, the greater the academic interest in the paper.

**Table 9.** Cluster 7.

| RO | Author | Links | Journal | Country | Citations | NIY |
|---|---|---|---|---|---|---|
| 1 | van der Waal et al. (2021) [165] | 101 | *Journal of Cleaner Production* | International | 25 | 25 |
| 2 | Johnsson et al. (2020) [168] | 67 | *Renewable and Sustainable Energy Reviews* | n.a. | 32 | 16 |
| 3 | Khan et al. (2021) [169] | 20 | *Business Strategy and the Environment* | n.a. | 15 | 15 |
| 4 | Ordonez-Ponce et Khare (2021) [170] | 100 | *Journal of Environmental Planning and Management* | - | 14 | 14 |
| 5 | Ike et al. (2019) [166] | 66 | *Journal of Cleaner Production* | Japan | 41 | 13.67 |
| 6 | Jan et al. (2021) [171] | 21 | *Sustainability* (Switzerland) | Islamic countries | 8 | 8 |
| 7 | ElAlfy et al. (2020) [172] | 109 | *Sustainable Development* | International | 14 | 7 |
| 8 | Calabrese et al. (2021) [173] | 108 | *Journal of Cleaner Production* | - | 6 | 6 |
| 9 | Szennay et al. (2019) [174] | 67 | *Resources* | n.a. | 18 | 6 |
| 10 | Warmate et al. (2021) [175] | 11 | *Business Strategy and the Environment* | International | 5 | 5 |
| 11 | Russell et al. (2018) [176] | 11 | *Sustainability* (Switzerland) | United Kingdom | 20 | 5 |
| 12 | Gerged and Almontaser (2021) [13] | 54 | *Resources Policy* | Libya | 3 | 3 |
| 13 | Diaz-Sarachaga (2021) [29] | 100 | *Corporate Social Responsibility and Environmental Management* | Spain | 2 | 2 |
| 14 | Lourenção et al. (2021) [177] | 35 | *World Review of Entrepreneurship, Management and Sustainable Development* | - | 2 | 2 |
| 15 | Lee and Kim (2021) [30] | 14 | *Social Indicators Research* | International | 2 | 2 |
| 16 | Elalfy et al. (2020) [178] | 121 | *Journal of Applied Accounting Research* | - | 4 | 2 |
| 17 | Vogel-Pöschl et al. (2020) [179] | 51 | *Zeitschrift fur Evaluation* | - | 2 | 1 |
| 18 | Caldana et al. (2021) [167] | 35 | *Benchmarking* | Brazil | 0 | 0 |
| 19 | Galleli et al. (2021) [17] | 34 | *Sustainability* (Switzerland) | Brazil | 0 | 0 |

RO: ranking order; NIY: normalized citations per year; Source: Scopus.

*-Cluster 8 (colored brown)—SDG reporting. Nature and orientation:* Eleven papers that analyzed that analyze the nature and orientation of SDG reporting by companies make up this cluster. These articles are connected with those belonging to clusters 3 and 7.

The articles with the highest number of links were those by Pzzi et al. (2020) and Izzo et al. (2020). With regards to the articles' impact, the paper with the highest number of citations, both in absolute and in relative terms, was that by Rosati and Faria (2019) [9]. The following papers with a higher academic impact were those by Pizzi et al. (2021) [180] and de Villiers et al. (2021) [181]. Conversely, there were two papers with no citations.

All of the articles belonging to this cluster were written by multiple authors. The author with the highest number of publications was Mukherjee, M., with two papers. Within this subtopic, the journal with more papers published was *Sustainability* (three papers), followed by the *Journal of Cleaner Production* (two papers). The first published article of this cluster dated from 2019, and the year with more publications was 2021, with seven papers published during that year.

Table 10 shows the papers belonging to this cluster, their journal, the number of links between papers, the country or region of study, and their impact or influence measured by the total number of citations and the average number of citations per year from the date of publication (NIY) [51]. It should be noted that the last column reflects the "acceleration" of the impact in time weighting. Thus, under equal conditions of the date of publication, the greater the NIY, the greater the academic interest in the paper.

**Table 10.** Cluster 8.

| RO | Author | TL | Journal | Country | Citations | NIY |
|----|--------|----|---------|---------|-----------|-----|
| 1 | Rosati and Faria (2019) [9] | 96 | *Journal of Cleaner Production* | International | 161 | 53.67 |
| 2 | Pizzi et al. (2020) [180] | 118 | *Journal of Cleaner Production* | n.a. | 73 | 36.5 |
| 3 | de Villiers et al. (2021) [181] | 65 | *Journal of Business Research* | n.a. | 20 | 20 |
| 4 | Izzo et al. (2020) [182] | 115 | *Sustainability* (Switzerland) | Italy | 27 | 13.5 |
| 5 | Di Vaio et al. (2021) [183] | 36 | *Maritime Policy and Management* | n.a. | 6 | 6 |
| 6 | Ghosh and Rajan (2019) [184] | 9 | *International Journal of Sustainable Development and World Ecology* | International | 16 | 5.33 |
| 7 | Gambetta et al. (2021) [7] | 43 | *Sustainability* (Switzerland) | Spain | 5 | 5 |
| 8 | Mukherjee and Wood (2021) [185] | 10 | *Sustainability* (Switzerland) | Vietnam, Indonesia, Malaysia, and the Philippines | 2 | 2 |
| 9 | Franco-Riquelme and Rubalcaba (2021) [186] | 30 | *Journal of Open Innovation: Technology, Market, and Complexity* | Spain | 1 | 1 |
| 10 | Nguyen and Ngo (2021) [187] | 39 | *Economic Research-Ekonomska Istrazivanja* | Vietnam | 0 | 0 |
| 11 | Boffa and Maffei (2021) [188] | 7 | *FME Transactions* | n.a. | 0 | 0 |

RO: ranking order; NIY: normalized citations per year; Source: Scopus.

*-Cluster 9 (colored pink)—SDGs and business strategies:* Six papers that analyzed SDGs and their relationship with business strategies, analyzing which strategies facilitate SDGs' implementation, make up this cluster.

The articles with the highest number of links were those by Mio et al. (2020). With regards to the articles' impact, the paper with the highest number of citations, both in absolute and in relative terms, was that by Mio et al. (2020) [39]. The following paper with a higher academic impact was that by El-Haddadeh et al. (2021) [189]. Conversely, the paper with the lower number of citations was that by van den Broek (2020) [190].

All the articles belonging to this cluster were written by multiple authors except for one, and all the authors had one published article about this subtopic. Within this subtopic, the journal with more papers published was *Sustainability* (two papers), while the other journals had one article each. The first published article of this cluster dated from 2019, and

the year with more publications was 2021, with three papers published during that year. Moreover, European countries were the most analyzed.

Table 11 shows the papers belonging to this cluster, their journal, the number of links between papers, the country or region of study, and their impact or influence measured by the total number of citations and the average number of citations per year from the date of publication (NIY) [51]. It should be noted that the last column reflects the "acceleration" of the impact in time weighting. Thus, under equal conditions of the date of publication, the greater the NIY, the greater the academic interest in the paper.

**Table 11.** Cluster 9.

| RO | Author | Links | Journal | Country | Citations | NIY |
|----|--------|-------|---------|---------|-----------|-----|
| 1 | Mio et al. (2020) [39] | 121 | *Business Strategy and the Environment* | n.a. | 55 | 27.5 |
| 2 | El-Haddadeh et al. (2021) [189] | 3 | *Journal of Business Research* | United Kingdom | 15 | 15 |
| 3 | Shereni (2019) [191] | 1 | *African Journal of Hospitality, Tourism and Leisure* | Sub-Saharan African countries | 7 | 2.33 |
| 4 | Jimenez et al. (2021) [192] | 50 | *Sustainability* (Switzerland) | n.a. | 2 | 2 |
| 5 | Camodeca and Almici (2021) [193] | 20 | *Sustainability* (Switzerland) | Italy | 2 | 2 |
| 6 | van den Broek (2020) [190] | 63 | *Corporate Communications* | French | 2 | 1 |

RO: ranking order; NIY: normalized citations per year; Source: Scopus.

Moreover, we conducted a co-occurrence analysis, which is based on the idea that "the relatedness of items is determined based on the number of documents in which they occur together" (vosViewer database). In this case, the unit of analysis is the keywords (considering all keywords). We established a minimum number of occurrences of a keyword (5) and, from the 1.148 keywords of our sample, 59 met these conditions. Figure 8 shows the results of the co-occurrence analysis. The most used keywords were: sustainable development (total link strength: 380), sustainable development goal (total link strength: 353), sustainable development goals (total link strength: 269), and sustainability (total link strength: 239). It is remarkable that "private sector" was only repeated 15 times and "business" 17, when they constitute the other fundamental point of the articles that we are analyzing. This suggests that the most specific keywords are not really being used to classify the papers, since it seems necessary to use some reference to the private sector as a keyword to differentiate the works that analyze the business sector from those that deal with the public sector or NGOs.

In our final sample, we found 5.975 cited sources, of which only 22 journals received more than 40 citations. Table 12 shows the ten journals that received the highest number of citations, as well as the number of citations per year. These numbers clearly reflect the importance of the *Journal of Cleaner Production* in the discussion of the role of companies in meeting the SDGs.

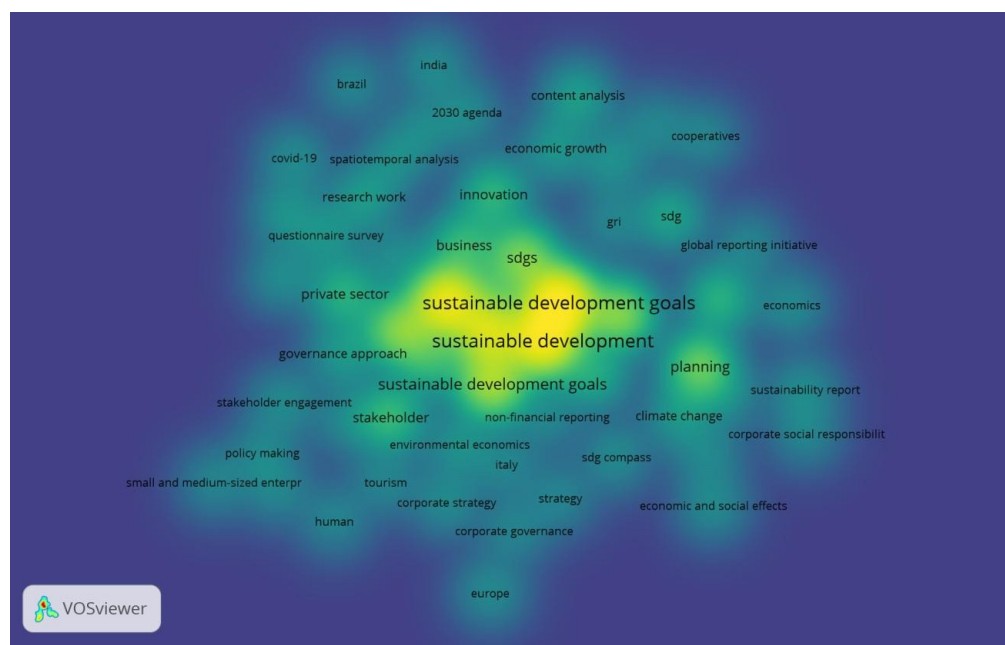

**Figure 8.** Keywords. Source: vosViewer and Scopus.

**Table 12.** Number of citations per journal.

| Journal | Citations | NIY |
|---|---|---|
| *Journal of Cleaner Production* | 700 | 116.67 |
| *Journal of Business Ethics* | 526 | 87.67 |
| *Sustainability* | 366 | 61 |
| *Corporate Social Responsibility and Environmental Management* | 272 | 45.33 |
| *Business Strategy and the environment* | 155 | 25.83 |
| *Sustainable Development* | 131 | 21.83 |
| *Academy of Management Review* | 75 | 12.5 |
| *Nature* | 57 | 9.5 |
| *Strategic Management Journal* | 51 | 8.5 |
| *Accounting, Auditing & Accountability Journal* | 49 | 8.17 |

NIY: normalized citations per year; Source: vosViewer and Scopus.

On the other hand, we found 17.826 cited authors, of which only 21 had been cited more than 40 times. Table 13 shows the ten authors who were cited more than 60 times, as well as the number of citations per year.

Finally, from the 13.967 cited references, only two had been cited more than 10 times (Table 14). This makes sense because, as we stated, this is a very current research topic, so the two most cited articles are among the oldest.

**Table 13.** Number of citations per author.

| Authors | Citations | NIY |
|---|---|---|
| Rosati, F. | 93 | 15.5 |
| Kolk, A. | 81 | 13.5 |
| Van Tulder, R. | 72 | 12 |
| Griggs, D. | 68 | 11.33 |
| García-Sánchez, I.M | 66 | 11 |
| Scheyvens, R. | 64 | 10.67 |
| Rockstrom, J. | 61 | 10.17 |
| Bebbington, J. | 60 | 10 |
| Schaltegger, S. | 59 | 9.83 |
| Unerman, J. | 59 | 9.83 |

NIY: normalized citations per year; Source: vosViewer and Scopus.

**Table 14.** Most cited references.

| Reference | Citations | NIY |
|---|---|---|
| Sullivan, K., Thomas, S., & Rosano, M. (2018). Using industrial ecology and strategic management concepts to pursue the Sustainable Development Goals. *Journal of Cleaner Production*, 174, 237–246. | 13 | 4.33 |
| Scheyvens, R., Banks, G., & Hughes, E. (2016). The private sector and the SDGs: The need to move beyond 'business as usual'. *Sustainable Development*, 24(6), 371–382. | 10 | 2 |

NIY: normalized citations per year; Source: vosViewer and Scopus.

## 5. Discussion

### 5.1. Main Characteristics of the Papers

In this section, we summarize the main characteristics of the papers under study. In addition to the issues analyzed so far, is interesting to expose the theories on which they have been based, the SDGs that they analyze, or the characteristics of the sample. It is interesting to analyze this information jointly, since issues are observed that provide relevant data regarding the status of existing research on the role that companies play in the development of the SDGs.

Much of the work obtained in this bibliographic review resorted to the theories that have been commonly used in CSR research to reinforce their work, as can be seen in Table 15. Moreover, the papers that used a theoretical framework mainly did so individually, although there were some works that combined several of these theories. Other papers based their research on the theoretical framework of the SDGs, but were not based on specific theories (e.g., [181,184]). It should be noted that, in the first cluster the most used theories were the stakeholder theory and the institutional theory. Without a doubt, the third cluster was the one that showed the greatest variety of theories, and it was also the cluster that presented a greater number of studies that based their framework on an existing theory.

**Table 15.** Theories used in the papers analyzed.

| Theory | Papers |
|---|---|
| Activity theory | Saz-Gil et al. (2020) [81] |
| Agency theory | Gambetta et al. (2021) [7]; Khaled et al. (2021) [14]; García-Meca and Martínez-Ferreiro (2021) [93]; García-Sánchez et al. (2019) [95]; Kazemikhasragh et al. (2021) [107]; Lassala et al. (2021) [149] |
| Continuity theory | Saz-Gil et al. (2020) [81] |
| Grounded theory | Jan et al. (2021) [171] |
| Impression management theory | García-Sánchez et al. (2020) [38] |
| Institutional theory | Rosati and Faria (2019) [9]; Gerged and Almontaser (2021) [13]; Galleli et al. (2021) [17]; van Zanten and van Tulder (2018) [34]; García-Sánchez et al. (2020) [36]; Hepner et al. (2021) [61]; Erin and Bamigboye (2021) [97]; Izzo et al. (2020) [101]; García-Sánchez et al. (2019) [103]; Ordonez-Ponce and Khare (2021) [170] |
| Legitimacy theory | Gambetta et al. (2021) [7]; Rosati and Faria(2019) [9]; García-Sánchez et al. (2020) [38]; Yu et al. (2020) [56]; Curtó-Pagès et al. (2021) [91]; García-Meca and Martínez-Ferreiro (2021) [93]; Izzo et al. (2020) [101]; Kazemikhasragh et al. (2021) [107]; De Luca et al. (2020) [116]; Yu and Kuo (2021) [126]; Lassala et al. (2021) [149]; Khan et al. (2021) [169]; ElAlfy et al. (2020) [172] |
| Natural resource-based view | Ilyas et al. (2020) [111] |
| Organizational identity theory | Liou and Rao-Nicholson (2021) [135] |
| Paradox theory | Vildåsen (2018) [64] |
| Resource-based view | Ordonez-Ponce et al. (2021) [18] |
| Signaling theory | Rosati and Faria(2019) [9]; Diaz-Sarachaga (2021) [29]; Khan et al. (2021) [169] |
| Social and environmental justice theory | Gutberlet (2021) [133] |
| Stakeholder theory | Gambetta et al. (2021) [7]; Rosati and Faria(2019) [9]; Diaz-Sarachaga (2021) [29]; Jonsdottir et al. (2021) [35]; Gunawan et al. (2020) [48]; Lopez (2020) [59]; García-Sánchez et al. (2019) [95]; Gallego-Sosa et al. (2021) [96]; Erin and Bamigboye (2021) [97]; Nishitani et al. (2021) [99]; Jun and Kim (2021) [104]; Modgil et al. (2020) [115]; Phan et al. (2020) [120]; Gallardo-Vázquez et al. (2021) [124]; Lassala et al. (2021) [149]; Jimenez et al. (2021) [192] |
| Temporality theory | van den Broek (2020) [190] |
| Theory of resource dependence | Gallego-Sosa et al. (2021) [96] |
| Upper Echelons theory | Gallego-Sosa et al. (2021) [96]; Ilyas et al. (2020) [111] |
| Value theory | Olofsson and Mark-Herbert (2020) [83] |
| Voluntary disclosure theory | Izzo et al. (2020) [182] |

On the other hand, a sign that the research on the subject is recent is that it can be seen that most of the studies approach the analysis from a generic point of view, focusing on the SDGs as a global concept. There is still not much specialized research on each of the SDGs. However, as shown in Table 16, some studies have conducted an analysis on a particular objective. Among these articles, we observed that the objective that received the most attention was 12 (Responsible consumption and production), followed by SDGs 8, 9, and 17. The only SDGs that had not been specifically analyzed were 2 and 16. The clusters that presented the most specialized studies on a specific SDG were 1, 5, and 6. In each of them, the most analyzed SDGs were also 12, 8, and 9

**Table 16.** Most cited references.

| SDG | Publications |
|---|---|
| 1 | Scheyvens and Hughes (2019) [72]; Gutberlet (2021) [133] |
| 2 | - |
| 3 | Hepner et al. (2021) [61]; Consolandi et al. (2020) [82] |
| 4 | Bello and Othman (2020) [63]; Mozas-Moral et al. (2020) [153]; Mozas-Moral et al. (2021) [2] |
| 5 | Hepner et al. (2021) [61]; Gutberlet (2021) [133]; Núñez et al. (2020) [150] |
| 6 | Hepner et al. (2021) [61] |
| 7 | Hepner et al. (2021) [61]; Modgil et al. (2020) [115] |
| 8 | Hepner et al. (2021) [61]; Modgil et al. (2020) [115]; Gutberlet (2021) [133]; Khalique et al. (2021) [144]; Matteucci (2020) [147]; Núñez et al. (2020) [150]; Mozas-Moral et al. (2020) [153]; Mozas-Moral et al. (2021) [2] |
| 9 | Hepner et al. (2021) [61]; Vildåsen (2018) [64]; Modgil et al. (2020) [115]; Nobrega et al. (2021) [127]; Mozas-Moral et al. (2020) [153]; Mozas-Moral et al. (2021) [2] |
| 10 | Núñez et al. (2020) [150] |
| 11 | Di Vaio and Varriale (2020) [94]; Modgil et al. (2020) [115]; Gutberlet (2021) [133] |
| 12 | Palakshappa and Dodds (2021) [55]; Hepner et al. (2021) [61]; Vildåsen (2018) [64]; Modgil et al. (2020) [115]; Gutberlet (2021) [133]; Matteucci (2020) [147]; Mozas-Moral et al. (2020) [153]; Mozas-Moral et al. (2021) [2]; Russell et al. (2018) [176] |
| 13 | Mozas-Moral et al. (2020) [153]; Mozas-Moral et al. (2021) [2] |
| 14 | Vildåsen (2018) [64] |
| 15 | Hepner et al. (2021) [61]; Mozas-Moral et al. (2020) [153]; Mozas-Moral et al. (2021) [2] |
| 16 | - |
| 17 | Hepner et al. (2021) [61]; Vildåsen (2018) [64]; Di Vaio and Varriale (2020) [94]; Matteucci (2020) [147]; Mozas-Moral et al. (2021) [2] |

The research on the role that companies play in the fulfillment of the SDGs is mainly empirical, although there have also been several studies that carried out literature reviews and approached the subject from a theoretical point of view (e.g., [152,169,192]). In those cases in which the analysis was carried out in a practical way, the most used methodology was content analysis (e.g., [7,123,164]). These works mainly analyzed the different types of business reports (non-financial reports, annual reports, or sustainability reports), and corporate websites.

At the business level, we observed what has been commented on for a long time in the academic literature. Most studies have focused on the role of large companies. The most common samples are listed firms, top companies, or multinationals, with SMEs being much less frequent in this research. From the sectoral point of view, there have not been many works that focused on a particular sector, but it was clearly appreciated that the most analyzed sector was tourism (e.g., [66,77]).

*5.2. Academic Impact of the Papers*

Regarding the academic impact of the papers, in four clusters, the paper with the highest relative impact (NIY) was also that with the highest number of citations (absolute impact). This was the case for cluster 1 (Scheyvens et al., 2016), cluster 6 (Mino et al., 2021), cluster 8 (Rosadi and Faria, 2019), and cluster 9 (Mio et al., 2020). It should be noted that two of these papers were very recent (2020 and 2021), and both were published in the same journal (*JCP*), which was also the journal with the highest number of citations in the sample. On the other hand, the article from Scheyvens et al. was the first published paper on this topic.

With regards to the remaining clusters, (2, 3, 4, 5, and 7), there was asymmetry between the relative and absolute impact. In all of these clusters, when comparing the papers with

the higher absolute impact and the ones with higher NIY, the former were those with more citations. However, to assess the actual research interest in a paper, it is necessary to consider the NIY, as the papers with a higher NIY received fewer total citations, but all of them had been published in 2021 and 2020. Therefore, this could have influenced their total citations. The NIY allows visualizing the papers addressing a "hot topic".

Most of the papers with higher academic impact were published in two of the analyzed years. This indicates that the topic is very attractive to researchers. The most impactful papers are those addressing the 2030 Agenda from a wide viewpoint, instead of focusing on a specific SDG. Likewise, most of the papers with a higher impact focused on an international sample, while some impactful papers analyzed a single country (Indonesia, Spain, Australia, Japan, Italy, and the UK).

### 5.3. Publication Opportunities

Based on the papers with a higher absolute and relative impact, we will try to offer some suggestions for future research. The papers with the highest absolute and relative impact belonged to Clusters 1, 3, and 8. The latter two addressed issues related to SDG reporting (determinants and nature), whereas the latter focused on how businesses address the SDGs. Most of them adopted an international perspective and a broad focus, without considering specific SDGs. Conversely, the papers belonging to clusters 2, 5, 7, and 9 had not been the subject of high research attention.

The fact that academics are interested in the topics addressed in clusters 1, 3, and 8 could indicate the direction to be followed by future studies, as such topics can be considered "hot topics" in which both journals and researchers are interested. The fact that cluster 8 was made up of only 11 articles and the work with the highest relative impact belonged to it reflects that this cluster provides academics interested in the 2030 Agenda a wide range of opportunities to contribute to this field.

## 6. Concluding Remarks

Considering the importance of the business sector in meeting the SDGs, this work aims to investigate the scope of the existing literature about the role that companies can play in contributing to the fulfillment of SDGs. A bibliometric analysis was carried out to research the papers on the relationship between business and the SDGs published from 2015 to 2021. With the aim of systemizing research on the role of companies in meeting the SDGs, we studied our sample and analyzed the authors, journals, countries, and the temporal evolution of this topic within the academic world.

Our final sample was composed of 196 papers that analyzed the role of business in achieving the SDGs. Most of them were published since 2019 (80%), reflecting that we are facing a young research issue. The presence of this topic in the literature has experienced remarkable growth in recent years, which demonstrates the relevance of and interest in this topic.

Moreover, the journal with the most papers published on this topic is *Sustainability*, with 50 documents throughout the studied six years, followed by the *Journal of Cleaner Production*, with 15 papers, and *Business Strategy and the Environment*, with six.

Many authors have shown interest in investigating the role that companies play in the implementation of the SDGs, but they have not yet been particularly fruitful. Very few (11 authors) have written more than three articles in this field of research. It should be noted that most of the articles that investigated this topic were written by several authors, with the number of works carried out by a single author being clearly lower. The author who has published the most articles on this subject is García-Sánchez, and Spain is the country with the highest number of publications.

In this analysis, we obtained 11 clusters, of which only 9 were really relevant as research topics. Among them, the articles were classified according to different criteria, from how they can implement the SDGs to the measures that companies must adopt for their evaluation. The most analyzed clusters were the first three, which made reference to

how businesses address the SDGs, the benefits arising from SDG engagement, and SDG reporting. On the other hand, the least analyzed cluster was cluster 9, which dealt with the subject of SDGs and business strategy. The first article published on this topic belonged to cluster 1.

As López-Concepción et al. noted, research on businesses' contribution to the SDGs is "unstructured and fragmented" [194] (p. 2); thus, our bibliometric analysis contributes to providing a reference frame of the state of the art of this research topic, which can orientate researchers in the development of future studies. However, this work is subject to some limitations. Firstly, we included papers from the Scopus database as a source of data collection, but the Web of Science or Google Scholar should also be considered to expand the study. For example, it could be considered that only those papers published in JCR-indexed journals were used to obtain a view of the publications with a higher acknowledged quality and impact. Secondly, we used VOSviewer to carry out source analysis, but future studies could employ an alternative instrument (e.g., PRISMA-statement, SCImat) and compare the results. Moreover, the co-occurrence of international collaboration networks could enrich this research.

However, despite the limitations mentioned above, the relevance of this work is notable when it comes to contributing to the academic literature and practice. The summarizing of the existing research on the role that companies play in complying with the SDGs provides knowledge about the real involvement that organizations have in this issue. In addition, the differentiation of various themes into clearly identified clusters can serve as a future line of research for all those who wish to delve deeper into each of the underlying themes related to the SDGs.

**Author Contributions:** The whole article is the result of a joint project and shared effort. Conceptualization, M.G.-R. and B.A.-G.; methodology, M.G.-R. and B.A.-G.; software, M.G.-R.; validation, M.G.-R. and B.A.-G.; formal analysis, M.G.-R., B.A.-G. and A.P.M.; investigation, M.G.-R., B.A.-G. and A.P.M.; resources, M.G.-R.; data curation, M.G.-R. and B.A.-G.; writing—original draft preparation, M.G.-R. and B.A.-G.; writing—review and editing, M.G.-R. and B.A.-G.; visualization, M.G.-R., B.A.-G. and A.P.M.; supervision, M.G.-R., B.A.-G. and A.P.M.; project administration, M.G.-R., B.A.-G. and A.P.M. All authors have read and agreed to the published version of the manuscript.

**Funding:** This research received no external funding.

**Institutional Review Board Statement:** Not applicable.

**Informed Consent Statement:** Not applicable.

**Data Availability Statement:** The data presented in this study are available on request from the corresponding author.

**Conflicts of Interest:** The authors declare no conflict of interest.

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
