# Peer review of "Businesses’ Role in the Fulfillment of the 2030 Agenda: A Bibliometric Analysis"

_sustainability, doi:10.3390/su14148754_

Round 1

Reviewer 1 Report

The article is interesting and I must congratulate the authors for their approach and the proposal they make of their research in this preliminary version of the manuscript.

However, there are quite a few important aspects that need to be improved. Synthetically:

(1) Authors should underpin their methodology by following PRISMA, a robust international standard. In its current version, the manuscript is flimsy.
For this methodology, the authors should follow https://www.prisma-statement.org/ and a reference could be found (among others) here: https://www.prisma-statement.org/PRISMAStatement/PRISMAEandE.aspx

(2) In addition, when evaluating the thematic clusters (tables 2-10), they must incorporate the impact (influence) that these articles have achieved, measuring the impact with two dimensions: an absolute one (total number of citations), and a relative one ( average number of citations per year from the date of publication). This is very important, since the information they now provide does not say much by itself; while the impact normalized by years does offer detailed information about the "trend accelerometer" in academic debate.

For the normalized impact per year (NIY), the Castelló-Sirvent (2022) reference is available here:

Castelló-Sirvent, F. (2022). A Fuzzy-Set Qualitative Comparative Analysis of Publications on the Fuzzy Sets Theory. Mathematics, 10(8), 1322.

(3) In line 441 et seq., the authors state that "the right keywords or the most specific ones are not really being used to classify the papers". The authors must carry out a detailed analysis (point by point) that allows endorsing this analysis. SciMAT software is suggested for a strategy map analysis of the literature, but the authors may develop adequate justification for their claim based on other robust methodologies if they wish. In any case, they must clearly establish the difference between previously analyzed co-occurrences and cluster analysis based on "all keywords" (not author keywords, but also including keywords-plus, added by the journal)

Here the central reference of this software:

Cobo, M. J., López‐Herrera, A. G., Herrera‐Viedma, E., & Herrera, F. (2012). SciMAT: A new science mapping analysis software tool. Journal of the American Society for Information Science and Technology, 63(8), 1609-1630.  

(4) Authors must clearly justify based on the literature why they include Web of Science (WoS) in its entirety. This is very important, given that otherwise, your research is not solid, as you are considering indices other than the central ones (SSCI and SCIE) and you are building on an intentional dispersion of low quality. They must develop sufficient bases to support their complete choice of the Wos database and not just JCR, together with the choice of Scopus (here not discussed by this reviewer). Otherwise, they will have to redo all the calculations feeding their analysis with both databases, that is, Scopus and only the JCR high impact WoS section (SSCI and SCIE).

(5) In the same way that I have indicated in point (2) above, the authors must illustrate their results tables 11, 12 and 13 with the calculation of normalized citations per year (NIY), understood as the average of all the articles of the journals referred to in table 11, average of all the articles of the authors referred to in table 12 and the articles indicated in table 13.

(6) Regarding these last three tables, the authors must extend their report to a minimum of 10 (top10) journals, authors and articles, respectively. Similarly, they should incorporate this threshold in their previous analysis of table 1 and figure 4. Bias around 7 the number reported should be reviewed.

(7) The authors must reinforce their analysis with an evaluation of the co-occurrence of international collaboration networks: first, for institutions and, second, for countries of the affiliations of the authors of said institutions signing the articles.

(8) The authors must make a significant effort to develop a section for the discussion of results that allows their contribution to be connected with the existing gap in the literature. This is decisive to consider the publication of the article. This literature section should highlight how the authors allow the academic debate to advance.

(9) The conclusion section should be improved based on the findings and the real implications of the research for authors and journal editors. In the current version of the manuscript, the authors remain on the surface. The work to be done must be thorough and of high quality.

Author Response

Manuscript ID: sustainability-1774868

BUSINESS CONTRIBUTION TO THE SUSTAINABLE DEVELOPMENT GOALS: A BIBLIOMETRIC ANALYSIS

Dear Editor,

We are very grateful for the comments from the reviewers. Thanks for the opportunity to revise our paper and improve its quality. We are very grateful for the valuable, constructive comments and suggestions; they were very helpful in transforming the paper into the current version.

We have followed all their recommendations and, therefore, we have elaborated further on the previous draft by making the suggested changes. All the revisions are highlighted in yellow within the manuscript.

We sincerely hope that you find the revised manuscript suitable for publication in the Sustainability. Once again, thank you very much for your assistance.

Kind Regards,

The authors

REVIEWER 1

The article is interesting and I must congratulate the authors for their approach and the proposal they make of their research in this preliminary version of the manuscript. However, there are quite a few important aspects that need to be improved.

Dear Reviewer 1,

We are extremely grateful for your valuable comments and helpful suggestions. We have followed all your recommendations and, therefore, we have elaborated further on the previous draft introducing changes, which are described below.

Synthetically:

1.Authors should underpin their methodology by following PRISMA, a robust international standard. In its current version, the manuscript is flimsy.
For this methodology, the authors should follow https://www.prisma-statement.org/ and a reference could be found (among others) here: https://www.prisma-statement.org/PRISMAStatement/PRISMAEandE.aspx

Thank very much for your suggestions. It is very interesting and useful. Unfortunately, we are not able to address the change of software at this moment as it would imply to rewrite all the paper. However, we have included this suggestion as an option for future research.

Thus, in Section 3.2 (Data analysis and procedure) we added the following paragragh:

“Although there are other instruments that can be used for doing literature reviews (e.g., PRISMA-statement, SciMAT), we chose VOSviewer because it has been broadly used in previous studies (Monteiro et al., 2021; Boar et al., 2021).”

Then, in the Conclusions section we added the following sentence: “Secondly, we have used VOSviewer to carry out the sources analysis, but future studies could employ an alternative instrument (e.j., PRISMA-statement, SCImat) and compare the results.”

2.In addition, when evaluating the thematic clusters (tables 2-10), they must incorporate the impact (influence) that these articles have achieved, measuring the impact with two dimensions: an absolute one (total number of citations), and a relative one ( average number of citations per year from the date of publication). This is very important, since the information they now provide does not say much by itself; while the impact normalized by years does offer detailed information about the "trend accelerometer" in academic debate. For the normalized impact per year (NIY), the Castelló-Sirvent (2022) reference is available here:

Castelló-Sirvent, F. (2022). A Fuzzy-Set Qualitative Comparative Analysis of Publications on the Fuzzy Sets Theory. Mathematics, 10(8), 1322.

Thank you very much fot this interesting and useful suggestion. Following it, we have included two additional columns in Tables 2-10 with the impact or influence of the papers measure. Moreover we explain before each table “Table 2-10 show the papers belonging to this cluster, their journal, the link between papers, the country or region of study and their impact or influence measured by the total number of citations and the average number of citations per year from the date of publication (Castelló-Sirvent, 2022).”

3.In line 441 et seq., the authors state that "the right keywords or the most specific ones are not really being used to classify the papers". The authors must carry out a detailed analysis (point by point) that allows endorsing this analysis. SciMAT software is suggested for a strategy map analysis of the literature, but the authors may develop adequate justification for their claim based on other robust methodologies if they wish. In any case, they must clearly establish the difference between previously analyzed co-occurrences and cluster analysis based on "all keywords" (not author keywords, but also including keywords-plus, added by the journal). Here the central reference of this software:

Cobo, M. J., López‐Herrera, A. G., Herrera‐Viedma, E., & Herrera, F. (2012). SciMAT: A new science mapping analysis software tool. Journal of the American Society for Information Science and Technology, 63(8), 1609-1630.

Thank you very much for your suggestion. Following it, we have included an explanation to this issue: “It is remarkable that “private sector” is only repeated 15 times and “business” 17, when it is the other fundamental point of the articles that we are analyzing. This suggests that the most specific keywords are not really being used to classify the papers, since it seems necessary to use some reference to the private sector as a keyword to differentiate the works that analyze the business sector with those that deal with the public sector or NGOs.”

Anyway, this analysis was carried out considering all the keywords, and we have replicated the analysis using author keywords to see if there was any significant difference, but we have obtained similar results. However, we consider to use your suggestion in future works.

  1. Authors must clearly justify based on the literature why they include Web of Science (WoS) in its entirety. This is very important, given that otherwise, your research is not solid, as you are considering indices other than the central ones (SSCI and SCIE) and you are building on an intentional dispersion of low quality. They must develop sufficient bases to support their complete choice of the Wos database and not just JCR, together with the choice of Scopus (here not discussed by this reviewer). Otherwise, they will have to redo all the calculations feeding their analysis with both databases, that is, Scopus and only the JCR high impact WoS section (SSCI and SCIE).

Thank you very much for your comment. We chose Scopuse “because includes a wide range of studies about this topic, more journals indexed than Web of Science and is a very common tool used for bibliometric studies [39], [40].” However, your proposal is very interesting and we considered as a future research avenue to compare our results with those obtained by applying a stricter criterio to select the papers (E.g. JCR): “For exaplame, it could be considered only those papers published in JCR indexed journals to obtain a view of the publications with a higher acknowledged quality and impact”.

5. In the same way that I have indicated in point (2) above, the authors must illustrate their results tables 11, 12 and 13 with the calculation of normalized citations per year (NIY), understood as the average of all the articles of the journals referred to in table 11, average of all the articles of the authors referred to in table 12 and the articles indicated in table 13.

Thank you very much for your comment. In line with your suggestion, we added a new column in Tables 12-14. We also wrote the paragraph regarding these Tables, including the reference to the citations per year: e.g.“Table 12 shows the ten journals that have received the highest number of citations as well as the number of citations per year”

6. Regarding these last three tables, the authors must extend their report to a minimum of 10 (top10) journals, authors and articles, respectively. Similarly, they should incorporate this threshold in their previous analysis of table 1 and figure 4. Bias around 7 the number reported should be reviewed.

Thank you very much for your comment. We extended the tables 11-12 in order to include the top 10 journals and authors, but we can we cannot expand table 13 since only two articles meet the characteristics mentioned in the text (to be cited more than 10 times).

Regarding table 1, we have mentioned the journals that publish more than 5 articles on that topic, that is why that table has 7 journals. And finally, regarding figure 4, we consider that it can be a bit confusing if we represent 10 lines of different colors, each referring to a journal.

  1. The authors must reinforce their analysis with an evaluation of the co-occurrence of international collaboration networks: first, for institutions and, second, for countries of the affiliations of the authors of said institutions signing the articles.

Thank you very much for your comment. We think it is a very interesting suggestion. Unfortunately, we are not available to make all the co-cocurrence analysis, but we have decided to carry out an analysis on the number of publications by institution and we suggest as future research to carry out a more in-depth analysis. “Finally, Table 7 shows the number of publications depending on the organization. To prepare the table, we have considered the most relevant organizations (those that have published 3 or more articles). Since the vast majority have published two (35 institutions) or one (111 organizations). As can be seen, the most productive universities are located in Europe. University of Salamanca is the only one with 5 publications. Although the first places belong mostly to European universities, it is worth highlighting the second place of University of Sao Paulo.”

Table 2. Documents by organization

Organization

TP

Universidad de Salamanca

5

Universidade de Sao Paulo

4

Universidad Rey Juan Carlos

4

University College London

4

Erasmus Universiteit Rotterdam

4

Parthenope University of Naples

4

Universidad de Santiago de Compostela

4

Rotterdam School of Management, Erasmus University

4

Universitat de Valencia

3

Universidad de Oviedo

3

Vaal University of Technology

3

Massey University

3

Technical University of Denmark

3

University of Waterloo

3

University of the Aegan

3

Syddansk University

3

Copenhagen Business School

3

Universita degli Studi di Roma Tor Vergata

3

Sant'Anna Scuola Universitaria Superiore Pisa

3

London South Bank University

3

University of South Australia

3

Kwame Nkrumah University of Science and Technology

3

Bartlett Faculty of the Built Environment

3

  1. The authors must make a significant effort to develop a section for the discussion of results that allows their contribution to be connected with the existing gap in the literature. This is decisive to consider the publication of the article. This literature section should highlight how the authors allow the academic debate to advance.

Thank you very much for your comment. We elaborate a section for the discussion as you recommend.

“5. Discussion

            In this section we summarizes the main characteristics of the papers under study. In addition to the issues analyzed so far, is interesting to expose the theories on which it has been based, the SDGs that they analyze or the characteristics of the sample. It is interesting to analyze this information jointly since issues are observed that provide relevant data on the status of existint research on the role that companies play in the development of the SDGs.

Much of the work obtained in this bibliographic review resort to the theories that have been commonly used in CSR research to reinforce their work as can be seen in table 15. Moreover, the papers that use a theoretical framework mainly do so individually, although there are some works that combine several of these theories. Other papers raise their research on the theoretical framework of the SDGs but are not based on specific theories (e.g. [55], [80]). It should be noted that in the first cluster the most used theories are the stakeholder theory and the institutional theory. And, without a doubt, the third cluster is the one that shows the greatest variety of theories, and it is also the cluster that presents a greater number of studies that base their framework on an existing theory.

Table 15. Theories used in the papers analyzed

Theory

Papers

Activity theory

Saz-Gil et al., 2020

Agency theory

García-Sánchez et al., 2020c; Gambetta et al., 2021; Kazemikhasragh et al., 2021; Khaled et al., 2021; Lassala et al., 2021;  García-Meca & Martínez-Ferreiro. 2021

Continuity theory

Saz-Gil et al., 2020

Grounded theory

Jan et al., 2021

Impression management theory

García-Sánchez et al., 2020

Institutional theory

van Zanten & van Tulder, 2018; Rosati & Faria, 2019; García-Sánchez et al., 2020a; 2020b; Izzo et al., 2020; Erin & Bamigboye, 2021; Gerged & Almontaser, 2021; Galleli et al., 2021; Hepner et al., 2021; Ordonez-Ponce & Khare, 2021

Legitimacy theory

Rosati & Faria, 2019; De Luca et al., 2020; ElAlfy et al.,2020; García-Sánchez et al., 2020; Izzo et al., 2020; Yu et al., 2020; Yu & Kuo, 2020; Curtó-Pagès et al., 2021; Gambetta et al., 2021; García-Meca & Martínez-Ferreiro, 2021; Khan et al., 2021; Kazemikhasrag et al., 2021; Lassala et al., 2021

Natural resource based view

Ilyas et al., 2020

Organizational identity theory

Liou & Rao-Nicholson, 2021

Paradox theory

Vildasen, 2018

Resource-based view

Ordóñez-Ponce et al., 2021

Signaling theory

Rosati & Faria, 2019; Díaz-Sarachaga, 2021; Khan et al., 2021

Social and environmental justice theory

Gutberlet, 2021

Stakeholder theory

Rosati & Faria, 2019; García Sánchez et al., 2020c; Gunawan et al., 2020; Lopez, 2020; Modgil et al., 2020; Phan et al., 2020; Díaz-Sarachaga, 2021; Erin & Bamigbaye, 2021; Gallardo-Vázquez et al., 2021; Gambetta et al., 2021; Jimenez et al., 2021; Jonsdottir et al., 2021; Lassala et al., 2021; Gallego-Sosa et al., 2021; Jun & Kim, 2021; Nishitani et al., 2021

Temporality theory

Van den Broek, 2020

Theory of resource dependence

Gallego-Sosa et al., 2021

Upper Echelons theory

Gallego-Sosa et al., 2020; Ilyas et al., 2020

Value theory

Olofsson & Mark-herbert, 2020

Voluntary disclosure theory

Izzo et al., 2020

On the other hand, a sign that the research on the subject is recent, it can be seen that most of the studies approach the analysis from a generic point of view, focusing on the SDGs as a global concept. There is still not much specialized research on each of the SDGs. However, as shown in Table 16, some studies do carry out an analysis on a particular objective. Among these articles, we observe that the objective that has received the most attention is 12 (Responsible consumption and production) followed by SDG 8, 9, 17. The only SDGs that are not specifically analyzed are 2 and 16. The clusters that present the most specialized studies in a specific SDG are 1, 5 and 6. And in each of them the most analyzed SDGs are also 12, 8 and 9

Table 16. Most cited references.

SDG

Publications

1

Scheyvens & Hughes, 2019; Gutberlet, 2021

2

-

3

Consolandi et al., 2020; Hepner et al., 2021

4

Bello & Othman, 2019; Mozas-Moral et al., 2020; Mozas-Moral et al., 2021

5

Núñez et al., 2020; Gutberlet, 2021; Hepner et al., 2021

6

Hepner et al., 2021

7

Modgil et al., 2020; Hepner et al., 2021

8

Modgil et al., 2020; Matteucci, 2020; Mozas-Moral et al., 2020; Núñez et al., 2020; Gutberlet, 2021; Hepner et al., 2021; Khalique et al., 2021; Mozas-Moral et al., 2021

9

Vildasen, 2018; Modgil et al., 2020; Mozas-Moral et al., 2020; Hepner et al., 2021; Nobrega et al., 2021; Mozas-Moral et al., 2021

10

Núñez et al., 2020

11

Di Vaio & Varriale, 2020; Modgil et al., 2020; Gutberlet, 2021

12

Russel et al., 2018; Vildasen, 2018; Modgil et al., 2020; Matteucci, 2020; Mozas-Moral et al., 2020; Gutberlet, 2021; Hepner et al., 2021; Palakshappa & Dodds, 2021; Mozas-Moral et al., 2021

13

Mozas-Moral et al., 2020; Mozas-Moral et al., 2021

14

Vildasen, 2018

15

Mozas-Moral et al., 2020; Hepner et al., 2021; Mozas-Moral et al., 2021

16

-

17

Vildasen, 2018; Di Vaio & Varriale, 2020; Matteucci, 2020; Hepner et al., 2021;  Mozas-Moral et al., 2021

 It can be seen that the research on the role that companies play in the fulfillment of the SDGs is mainly empirical, although there are also several studies that carry out literature reviews and approach the subject from a theoretical point of view (e.g., [66], [65], [75]). In those cases in which the analysis is carried out in a practical way, the most used methodology is content analysis (e.g. [84], [67], [94]). Mainly these works analyze the different types of business reports (non-financial reports, annual report or sustainability reports), and corporate websites.

 At the business level, we observe what has been commented for a long time in the academic literature. Most studies focus on the role of large companies. The most common samples are listed firms, top companies or multinationals with SMEs being much less frequent in this research. From the sectoral point of view, there are not many works that focus on a particular sector (e.g. [49], [133]), but it is clearly appreciated that the most analyzed sector is the tourism (e.g. [79], [74]).”

  1. The conclusion section should be improved based on the findings and the real implications of the research for authors and journal editors. In the current version of the manuscript, the authors remain on the surface. The work to be done must be thorough and of high quality.

Thank you for your comment. We have expanded the conclusion section adding the following paragraphs: “The most analyzed clusters are the first three, which make reference to how business address the SDGs, the benefits arising from SDG engagement and SDG reporting. On the other hand, the least analyzed cluster is the number nine, which deals with the subject of SDGs and bussines strategie. The first article published on this topic belongs to cluster 1.”.

“For example, it could be considered only those papers published in JCR indexed journals to obtain a view of the publications with a higher acknowledged quality and impact. Secondly, we have used VOSviewer to carry out the sources analysis, but future studies could employ an alternative instrument (e.j., PRISMA-statement, SCImat) and compare the results. Moreover, the co-occurrence of international collaboration networks could enrich this research.

However, and despite the limitations recently mentioned, we the relevance of this work when it comes to contributing to the academic literature and practice. The fact of summarizing the existing research on the role that companies play in complying with the SDGs provides knowledge about the real involvement that organizations have with this issue. In addition, the fact of differentiating various themes into clearly identified clusters can serve as future lines of research for all those who wish to delve deeper into each of the underlying themes related to the SDGs.”

Reviewer 2 Report

Dears authors,

Thank you for your research. 

In order to get the paper published, please consider the following:

1. Complete the FirstName, LastName, Affiliation, e-mail, etc for each author.

2. The abstract must show what is your contribution to this research. 

3. Your research methodology is widely used and your contribution is reduced. You must cite papers with the same research methodology and what is your contribution.

4. There must be an exact correspondence between references and citations in the article. There are titles in references that are not cited in the text, they only appear in the clusters identified in VOSViewer.

5. Please, complete the following section: Author Contribution, Funding, Data Availability Statement, Acknowledgments, Conflicts of Interest.

Author Response

Dears authors,

Thank you for your research. 

In order to get the paper published, please consider the following:

Dear Reviewer 2,

We are extremely grateful for your suggestions. We have followed all your recommendations and, therefore, we have elaborated further on the previous draft introducing changes, which are described below.

  • Complete the FirstName, LastName, Affiliation, e-mail, etc for each author.

Thank you for your comment and sorry for the mistake. We have already covered the corresponding data in the indicated place

  • The abstract must show what is your contribution to this research. 

Thanks for your comments. We included the following sentence in the abstract: “Moreover, it contributes to provide a reference frame of the state-of-art of this research topic and which can orientate researchers in the development of future studies”

  • Your research methodology is widely used and your contribution is reduced. You must cite papers with the same research methodology and what is your contribution.

Thanks for your comments. We included a discussion section in which we further developed these issues. Moreover, we add some cites of papers with the same research methodology as Monteiro et al., 2021 and Boar et al., 2021.

To clarify our contribution we added the following sentence in the introduction section: “we provide a systematization on the extant research on this subject that allows to identify knowledge flows, active research topics and lead authors, among other issues. Thus, this study’s findings depict the current status of research on the role of business in the fulfillment of SDGs and provide a reference frame that could guide researchers regarding the direction of future studies on this subject.”.

And the next sencente in the conclusions “However, and despite the limitations recently mentioned, we the relevance of this work when it comes to contributing to the academic literature and practice. The fact of summarizing the existing research on the role that companies play in complying with the SDGs provides knowledge about the real involvement that organizations have with this issue. In addition, the fact of differentiating various themes into clearly identified clusters can serve as future lines of research for all those who wish to delve deeper into each of the underlying themes related to the SDGs.

5.Discussion section

“In this section we summarizes the main characteristics of the papers under study. In addition to the issues analyzed so far, is interesting to expose the theories on which it has been based, the SDGs that they analyze or the characteristics of the sample. It is interesting to analyze this information jointly since issues are observed that provide relevant data on the status of existint research on the role that companies play in the development of the SDGs.

Much of the work obtained in this bibliographic review resort to the theories that have been commonly used in CSR research to reinforce their work as can be seen in table 15. Moreover, the papers that use a theoretical framework mainly do so individually, although there are some works that combine several of these theories. Other papers raise their research on the theoretical framework of the SDGs but are not based on specific theories (e.g. [55], [80]). It should be noted that in the first cluster the most used theories are the stakeholder theory and the institutional theory. And, without a doubt, the third cluster is the one that shows the greatest variety of theories, and it is also the cluster that presents a greater number of studies that base their framework on an existing theory.

Table 15. Theories used in the papers analyzed

Theory

Papers

Activity theory

Saz-Gil et al., 2020

Agency theory

García-Sánchez et al., 2020c; Gambetta et al., 2021; Kazemikhasragh et al., 2021; Khaled et al., 2021; Lassala et al., 2021;  García-Meca & Martínez-Ferreiro. 2021

Continuity theory

Saz-Gil et al., 2020

Grounded theory

Jan et al., 2021

Impression management theory

García-Sánchez et al., 2020

Institutional theory

van Zanten & van Tulder, 2018; Rosati & Faria, 2019; García-Sánchez et al., 2020a; 2020b; Izzo et al., 2020; Erin & Bamigboye, 2021; Gerged & Almontaser, 2021; Galleli et al., 2021; Hepner et al., 2021; Ordonez-Ponce & Khare, 2021

Legitimacy theory

Rosati & Faria, 2019; De Luca et al., 2020; ElAlfy et al.,2020; García-Sánchez et al., 2020; Izzo et al., 2020; Yu et al., 2020; Yu & Kuo, 2020; Curtó-Pagès et al., 2021; Gambetta et al., 2021; García-Meca & Martínez-Ferreiro, 2021; Khan et al., 2021; Kazemikhasrag et al., 2021; Lassala et al., 2021

Natural resource based view

Ilyas et al., 2020

Organizational identity theory

Liou & Rao-Nicholson, 2021

Paradox theory

Vildasen, 2018

Resource-based view

Ordóñez-Ponce et al., 2021

Signaling theory

Rosati & Faria, 2019; Díaz-Sarachaga, 2021; Khan et al., 2021

Social and environmental justice theory

Gutberlet, 2021

Stakeholder theory

Rosati & Faria, 2019; García Sánchez et al., 2020c; Gunawan et al., 2020; Lopez, 2020; Modgil et al., 2020; Phan et al., 2020; Díaz-Sarachaga, 2021; Erin & Bamigbaye, 2021; Gallardo-Vázquez et al., 2021; Gambetta et al., 2021; Jimenez et al., 2021; Jonsdottir et al., 2021; Lassala et al., 2021; Gallego-Sosa et al., 2021; Jun & Kim, 2021; Nishitani et al., 2021

Temporality theory

Van den Broek, 2020

Theory of resource dependence

Gallego-Sosa et al., 2021

Upper Echelons theory

Gallego-Sosa et al., 2020; Ilyas et al., 2020

Value theory

Olofsson & Mark-herbert, 2020

Voluntary disclosure theory

Izzo et al., 2020

On the other hand, a sign that the research on the subject is recent, it can be seen that most of the studies approach the analysis from a generic point of view, focusing on the SDGs as a global concept. There is still not much specialized research on each of the SDGs. However, as shown in Table 16, some studies do carry out an analysis on a particular objective. Among these articles, we observe that the objective that has received the most attention is 12 (Responsible consumption and production) followed by SDG 8, 9, 17. The only SDGs that are not specifically analyzed are 2 and 16. The clusters that present the most specialized studies in a specific SDG are 1, 5 and 6. And in each of them the most analyzed SDGs are also 12, 8 and 9

Table 16. Most cited references.

SDG

Publications

1

Scheyvens & Hughes, 2019; Gutberlet, 2021

2

-

3

Consolandi et al., 2020; Hepner et al., 2021

4

Bello & Othman, 2019; Mozas-Moral et al., 2020; Mozas-Moral et al., 2021

5

Núñez et al., 2020; Gutberlet, 2021; Hepner et al., 2021

6

Hepner et al., 2021

7

Modgil et al., 2020; Hepner et al., 2021

8

Modgil et al., 2020; Matteucci, 2020; Mozas-Moral et al., 2020; Núñez et al., 2020; Gutberlet, 2021; Hepner et al., 2021; Khalique et al., 2021; Mozas-Moral et al., 2021

9

Vildasen, 2018; Modgil et al., 2020; Mozas-Moral et al., 2020; Hepner et al., 2021; Nobrega et al., 2021; Mozas-Moral et al., 2021

10

Núñez et al., 2020

11

Di Vaio & Varriale, 2020; Modgil et al., 2020; Gutberlet, 2021

12

Russel et al., 2018; Vildasen, 2018; Modgil et al., 2020; Matteucci, 2020; Mozas-Moral et al., 2020; Gutberlet, 2021; Hepner et al., 2021; Palakshappa & Dodds, 2021; Mozas-Moral et al., 2021

13

Mozas-Moral et al., 2020; Mozas-Moral et al., 2021

14

Vildasen, 2018

15

Mozas-Moral et al., 2020; Hepner et al., 2021; Mozas-Moral et al., 2021

16

-

17

Vildasen, 2018; Di Vaio & Varriale, 2020; Matteucci, 2020; Hepner et al., 2021;  Mozas-Moral et al., 2021

 It can be seen that the research on the role that companies play in the fulfillment of the SDGs is mainly empirical, although there are also several studies that carry out literature reviews and approach the subject from a theoretical point of view (e.g., [66], [65], [75]). In those cases in which the analysis is carried out in a practical way, the most used methodology is content analysis (e.g. [84], [67], [94]). Mainly these works analyze the different types of business reports (non-financial reports, annual report or sustainability reports), and corporate websites.

 At the business level, we observe what has been commented for a long time in the academic literature. Most studies focus on the role of large companies. The most common samples are listed firms, top companies or multinationals with SMEs being much less frequent in this research. From the sectoral point of view, there are not many works that focus on a particular sector (e.g. [49], [133]), but it is clearly appreciated that the most analyzed sector is the tourism (e.g. [79], [74]).“

  • There must be an exact correspondence between references and citations in the article. There are titles in references that are not cited in the text, they only appear in the clusters identified in VOSViewer.

Thanks for your comments.

Thanks for your comments. This happens because there are conclusions that we make in a generic way referring to the clusters and since they are already mentioned in the tables, we do not repeat the references in the text. In any case, we have revised the references and citations of the document again to avoid errors.

  • Please, complete the following section: Author Contribution, Funding, Data Availability Statement, Acknowledgments, Conflicts of Interest.

Sorry for the mistake. We have already covered the corresponding data in the indicated place

Author Contributions: The whole article is the result of a joint project and shared effort. Conceptualization, M.G.-R. and B.A.-G.; methodology, M.G.-R. and B.A.-G.; software, M.G.-R.; validation, M.G.-R. and B.A.-G.; formal analysis, M.G.-R., B.A.-G. and A.P.M; investigation, M.G.-R., B.A.-G. and A.P.M; resources, M.G.-R.; data curation, M.G.-R. and B.A.-G.; writing—original draft preparation, M.G.-R. and B.A.-G.; writing—review and editing, M.G.-R. and B.A.-G.; visualization, M.G.-R., B.A.-G. and A.P.M; supervision, M.G.-R., B.A.-G. and A.P.M; project administration, M.G.-R., B.A.-G. and A.P.M. All authors have read and agreed to the published version of the manuscript.

Funding: This research received no external

Data Availability Statement: The data presented in this study are available on request from the corresponding author.

Conflicts of Interest: The authors declare no conflict of interest.

Round 2

Author Response

I want to convey to the authors my congratulations for the changes made to the manuscript. The current version has improved a lot in quality, however, some important details are still missing that must be taken into account. Synthetically:

Dear Reviewer 1,

We are extremely grateful for your suggestions. We want to thank you for your work in revising this paper and for your efforts to achieve better quality. Once again, we have followed al your recommendations and, therefore, we have elaborated further on the previous draft introducing changes, which are described below.

(1) The relative indicator of acceleration of the impact in time weighting (Normalized Impact per Year: NIY) should be explained more clearly, particularly its analytical implications: the higher the NIY, under equal conditions of the date of publication of the article, greater academic interest in that article-track-journal-production of a given country; for recent productions (eg 4-5 years ago), NIY coefficient allows visualizing a heat map of the interest of the academy, in a time acceleration perspective, offering researchers clues about publication opportunities.

Thank you for your comments. We have include a brief explanation fo the meaning of this column before each table: “It should be noticed that the last column reflects the “acceleration” of the impact in time weighting. Thus, under equal conditions of the date of publication, greater NIY, greater academic interest in the paper”

(2) In the discussion section, the authors should extract better links between the evidence that supports their results.

A good example is that in Tables 3 and following, whose order (column RO) established by the authors based on the number of links of the articles, that is, the number of articles included in the respective reference sections, should be discussed and compared for each cluster with the two columns on the right (citations and NIY) and extract lessons learned from the empirical evidence found: there is no correlation between citations, NIY (accelerometer of influence and impact) or number of articles. In this sense, they should either justify the reason why they order (column RO) the results of the clusters (Tables 3 and following) based on links in the article (not relevant according to their discussion section in the current writing of the manuscript) and they do not do so by absolute impact (citations column) or relative impact considering the temporal acceleration variable of the impact, according to the weighting of the absolute impact, according to the time elapsed after the publication of the article (NIY).

Another option is for the authors to change the ordering of these tables and for the RO column to be ordered based on absolute impact (citations) or relative impact (NIY).

Thank you for your comments. Following your suggestion, the Tables have been modified and papers are ordered based on their relative impact.

However, it is necessary to improve the discussion section, since in its current version the manuscript fails to highlight the existing asymmetry between both impacts (citations and NIY), and this for each cluster.

Then we included a comment about this issue in the Discussion section: “In four clusters the paper with the highest relative impact (NIY) is also the one with the highest number of citations (absolute impact). This is the case of cluster 1 (Scheyvens et al., 2016), cluster 6 (Mino et al., 2021), cluster 8 (Rosati and Faria, 2019) and cluster 9 (Mio et al., 2020). It should be notice that two of these papers are very recent (2020 and 2021) and both are published in the same journal (Journal of Cleaner Production), which is also the journal with the highest number of citations in the sample. On the other hand, the article from Scheyvens et al. is the first published paper on this topic.

As regards the remaining clusters, (2, 3, 4, 5, and 7), there are an asymmetry between relative and absolute impact. In all of these clusters when comparing the papers with the higher absolute impact and the ones with higher NIY, the former are those with more citations. However to assess the actual research interest in a paper is necessary to consider the NIY as the papers with a higher NIY have received less total citations but all of them have been published in 2021 and 2020. Therefore this fact can be influenced their total citations. The NIY allows visualizing those papers that address a “hot topic”.”

Regardless of the above, the creation of a subsection dedicated to "publication opportunities", since it would allow checking the contributions of the manuscript to the literature and, especially, it would offer clear information on its practical implications, both for academics to guide their future research and for journal editors to articulate their strategy for publishing and launching special issues.

Thanks you for your comments. Following your suggestion we include a new subsection:

“5.3 Publications opportunities

Based on the papers with a higher absolute and relative impact, we will try to offer some suggestions for future research. The papers with the highest absolute and relative impact belong to Clusters 1, 3, and 8. The two latter address issues related to SDG reporting (determinants and nature) whereas the latter focuses on how businesses address the SDGs. Most of them adopt an international perspective and a broad focus without considering specific SDGs. Conversely, the papers belonging to clusters 2, 5, 7, and 9 do not have been subject of high research attention.

The fact that academics are interested in the topics addressed in clusters 1, 3, and 8 could sign the direction to be followed by future studies as such topics can be considered “hot topic” on which both journals and researchers are interested. Besides, the fact that the cluster 8 is made up by only 11 articles and the work with the highest relative impact belong to it reflect that this cluster provides academics interested in the 2030 Agenda a wide range of opportunities to contribute to this field.”

(3) Linking with the above, the discussion section should delve into the gaps detected within the academic discourse, and the analysis of the research agenda; In this way, the article must be articulated as a powerful tool that is based on the analysis and discussion of the results, particularly based on the differences between citations and NIY in its different variants to offer researchers and journal editors a map of publication opportunities, as it finds and highlights new trends that are “accelerating” in the last few years. Only in this way could the article have practical utility and help guide/advise future research in the mainstream context of the field of study that the authors are analyzing.

Thank you for your comment. We try to improve the Discussion section including the following paragraphs:

“5.2 Academic impact of the papers

Regarding the academic impact of the papers, in four clusters the paper with the highest relative impact (NIY) is also the one with the highest number of citations (absolute impact). This is the case of cluster 1 (Scheyvens et al., 2016), cluster 6 (Mino et al.,2021), cluster 8 (Rosadi and Faria, 2019) and cluster 9 (Mio et al., 2020). It should be notice that two of these papers are very recent (2020 and 2021) and both are published in the same journal (JCP), which is also the journal with the highest number of citations in the sample. On the other hand, the article from Scheyvens et al is the first published paper on this topic.

As regards the remaining clusters, (2, 3, 4, 5, and 7), there are an asymmetry between relative and absolute impact. In all of these clusters when comparing the papers with the higher absolute impact and the ones with higher NIY, the former are those with more citations. However to assess the actual research interest in a paper is necessary to consider the NIY as the papers with a higher NIY have received less total citations but all of them have been published in 2021 and 2020. Therefore this fact can be influenced their total citations. The NIY allows visualizing those papers that address a “hot topic”.

Most of the papers with higher academic impact have been published in the two analyzed years. This fact indicates that the topic is very attractive for researches. Besides, the most impactful papers are those that address the 2030 Agenda from a wide viewpoint instead of focusing on an specific SDG. Likewise most of the papers with a higher impact are focused on an international sample, while some impactful papers analyze a single country (Indonesia, Spain, Australia, Japan, Italy and the UK).

5.3 Publications opportunities

Based on the papers with a higher absolute and relative impact, we will try to offer some suggestions for future research. The papers with the highest absolute and relative impact belong to Clusters 1, 3, and 8. The two latter address issues related to SDG reporting (determinants and nature) whereas the latter focuses on how businesses address the SDGs. Most of them adopt an international perspective and a broad focus without considering specific SDGs. Conversely, the papers belonging to clusters 2, 5, 7, and 9 do not have been subject of high research attention.

The fact that academics are interested in the topics addressed in clusters 1, 3, and 8 could sign the direction to be followed by future studies as such topics can be considered “hot topic” on which both journals and researchers are interested. Besides, the fact that the cluster 8 is made up by only 11 articles and the work with the highest relative impact belong to it reflect that this cluster provides academics interested in the 2030 Agenda a wide range of opportunities to contribute to this field.”

(4) There are multiple references that are not correctly formatted following the Sustainbaility author guidelines. For more information, you can review this link: https://www.mdpi.com/journal/sustainability/instructions). Without exhaustive intention, some examples are indicated:

(4.1.) Page 18: Mansell et al (2020), Mansell et al (2020), Jiménez et al (2021), MozasMoral et al (2021)

(4.2.) Page 19: Elalfy et al (2020), Vogel-Pöschl et al (2020), Caldana et al (2021)

(4.3.) Page 21: Shereni (2019)

(4.4.) Page 24: all references included in Table 15

(4.5.) Page 25: all references included in Table 16

(4.6.) Lines 331, 352, 376, 394, 413, 432, 451, 467, 483. Regardless of these examples, authors should conduct an in-depth review of the entire manuscript.

Thank you for your comment and sorry for the mistake. We have already review all the references.

(5) Finally, multiple errata can be observed in the manuscript that must be corrected before final approval. Without being exhaustive, some examples are indicated:

(5.1.) lines 333, “NIY: normalized citations per year” should read “NIY: normalized citations per year”

(5.2.) Tables 15 and 16 must follow the table format established by the journal (which the authors do use in the previous 14 tables). For more information, you can check this link: https://www.mdpi.com/journal/sustainability/instructions).

Regardless of these examples, authors should conduct an in-depth review of the entire manuscript.

Thank you for your comment and sorry for the mistake. We have already review all the work and correct all detected errors.

(6) The source of all tables refers to “vosViewer and Scopus”. However, the source is the Scopus database. VosViewer is the software that has been used and does not constitute a data source. The authors must review this indication at the bottom of the table and to understand the meaning of the source they must think that the vosViewer software is not applicable, nor is the Microsoft Excel software applicable to calculate the NIY by dividing citations by the difference in years that exists between the year of preparation of the manuscript (2022) and the year of publication of each article.

Thank you for your comment and sorry for the mistake. We have already change the source of the tables.

With the completion of all the suggested changes, the article will be re-evaluated and, in the absence of additional questions that may appear along the way, it would have  the favorable evaluation of this reviewer for publication in Sustainability

Reviewer 2 Report

Congratulation!

Author Response

Dear Reviewer 2,

We are extremely grateful for your suggestions. We want to thank you for your work in revising this paper and for your efforts to achieve better quality.

Round 3

Reviewer 1 Report

I want to congratulate the authors for their efforts throughout the review process. In my opinion, the manuscript in its current version is ready for publication.

Kind regards.

Reviewer.